# BIG LEARNING

## ABSTRACT

Recent advances in foundation models reveal a promising direction for deep learning, with the roadmap steadily moving from big data to big models/neural-nets to the presented big learning. Specifically, the big learning exhaustively exploits the information inherent in its large-scale *complete/incomplete* training data, by simultaneously modeling many/all joint, conditional, and marginal data distributions across potentially diverse domains, with one universal foundation model. We reveal that the big learning principle ($i$) underlies most foundation models, ($ii$) is equipped with extraordinary flexibilities for complete/incomplete training data and various data generative tasks, ($iii$) potentially delivers all joint, conditional, and marginal data sampling capabilities with one universal model, and ($iv$) is a new dimension for upgrading conventional machine learning paradigms. We leverage the big learning principle to upgrade the generative adversarial nets (in this paper), the expectation-maximization algorithm (in the supplementary), and the variational auto-encoders (in the supplementary) to their big-learning variants, with diverse experiments conducted to justify its effectiveness.

## 1 INTRODUCTION

AI is undergoing a paradigm shift with the rise of foundation models [4; 53], *e.g.,* BERT [44], GPTs [6; 37; 35; 36], the MAE [20], DALL-Es [39; 40], Imagen [42], Stable Diffusion [41], UniDiffuser [2], *etc.* Foundation models, often based on pretraining on broad data at scale, have demonstrated amazing modeling capabilities across diverse domains with impressive robustness [44], adaptability [20], and generalization [39]. Therefore, they are rapidly being integrated into real-world AI systems, *e.g.,* BERT into Google search, Codex [7] into GitHub's Copilot, ChatGPT/GPT-4 into Microsoft windows [35; 36], *etc.*

Despite the impressive capabilities and characteristics of foundation models, a unified theoretical framework justifying their great successes remains missing [4; 53], which is believed crucial for their further improvements and is likely a milestone for the foundation model community [45]. The presented big learning is considered as one step towards addressing that challenge.

Below we first summarize two main reasons for the successes of foundation models, base on which we then unify most training objectives of foundation models, from the generative perspective, to reveal their underlying principle, *i.e.,* the big learning.

By referring to [4; 53], we attribute the successes of foundation models to the following two properties of their large-scale pretraining.

1. **Data comprehensiveness.** Foundation models are often pretrained with massive data with great diversity. Often collected with minimal human interventions, these pretraining data may be comprehensively consistent with the "true" data distribution that underlies both training/pretraining and test/finetuning phases, leading to a narrowed phase gap *from the data perspective* and, therefore, serving as one reason for the generalization and robustness of foundation models.

2. **Task comprehensiveness.** Foundation models are pretrained in a massive multitasking manner on a wealth of *data tasks*; *e.g.,* both masked language modeling (MLM) and causal LM (CLM) leverage one universal model to simultaneously model many conditional data distributions (see Section 3). Such massive-task pretraining shows foundation models comprehensive task experience, which narrows the training-test/pretraining-finetuning gap *from the task perspective* (it's likely the downstream task resembles a pretraining one).

Inspired by existing foundation models succeeding from their comprehensive pretraining data and tasks, we propose to enhance both comprehensiveness to the extreme with the presented big learning. Specifically, the big learning leverages a universal foundation model to simultaneously model *many/all* joint, conditional, and marginal data distributions across potentially diverse domains, manifested as a "big" *generative*[1] learning task that exhaustively exploits the data information from many/all perspectives.

Our contributions are summarized as follows.

- We propose the big learning to unify most training objectives of foundation models within one learning framework.
- We reveal that the big learning can be leveraged to deliver *many/all* joint, conditional, and marginal data sampling capabilities with one universal foundation model. Those capabilities, in general settings, can manifest as classification, generation, completion/in-painting, *etc.*
- We leverage the big learning principle to upgrade the conventional generative adversarial net (GAN) into its big-learning variant termed the BigLearn-GAN, which is a novel adversarially-trained foundation model.
- We empirically demonstrate that big learning ($i$) is feasible, ($ii$) delivers good model generalization, and ($iii$) can serve as a better strategy for finetuning foundation models.

## 2 PRELIMINARY

**Foundation models.** Taking shape in natural language processing (NLP), foundation models have drastically changed the research and practice of AI [4; 53]. BERT [44] and GPT series [38; 6] significantly accelerate the development of NLP, while models like DALL-Es [39; 40], Stable Diffusion [41], and UniDiffuser [2] effectively promote interdisciplinary research among different research fields, initiating a new revolution of AI-Generated Content (AIGC).

Most existing foundation models are pretrained with ($i$) masked LM (or masked auto-encoding; like BERT and MAE), ($ii$) causal/auto-regressive LM (like GPTs and DALL-E), and ($iii$) permutation LM (like XLNET [52]). See Table 1 for details. We will demonstrate in Section 3 that these pretraining methods are all special cases of the proposed big learning, which, accordingly, serves as a unified theoretical framework that reveals one underlying principle of foundation models.

**Transformers and Vision Transformers (ViTs).** Based on the flexible self-attention mechanism [47], Transformers have been serving as the de facto model architecture for foundation models. Often Transformers take as input an $L$-length sequence of discrete tokens $\boldsymbol{x} \in \mathbb{Z}^L$ and output the corresponding $D$-dimensional embedding $\boldsymbol{h} \in \mathbb{R}^{L \times D}$, with the self-attention mechanism flexibly customized (among the $L$ locations) to implement masked/causal/permutation LM. ViTs [14] are Transformers modified for modeling continuous image patches. Despite their high model capacity and flexible modeling capabilities, Transformers/ViTs are well-known to be over-parameterized and data/information hungry [29; 18; 49]; we will reveal that those properties of Transformers/ViTs exactly matches the big learning.

**Multi-mode training objectives.** Two well-known multi-mode training objectives are ($i$) the cross-entropy loss, often used in maximum likelihood learning with *discrete* categorical observations, and ($ii$) the GAN loss [15] for adversarial learning on *continuous* observations, as detailed below.

1. **The cross-entropy loss.** Given history observations $\boldsymbol{x}$ and the current word $y$ sampled from the underlying data distribution $q(\boldsymbol{x}, y)$, and a model $p_{\boldsymbol{\theta}}(y|\boldsymbol{x})$ modeling the categorical distribution of $y$ given $\boldsymbol{x}$, the cross-entropy loss is identical to

$$\mathbb{E}_{q(\boldsymbol{x}, y)}[-\log p_{\boldsymbol{\theta}}(y|\boldsymbol{x})] \propto \mathrm{KL}[q(\boldsymbol{x}, y)||p_{\boldsymbol{\theta}}(y|\boldsymbol{x})q(\boldsymbol{x})], \tag{1}$$

   where the optimal $p_{\boldsymbol{\theta}^*}(y|\boldsymbol{x}) = q(y|\boldsymbol{x})$. Note the categorical modeling of $p_{\boldsymbol{\theta}}(y|\boldsymbol{x})$ can model multiple modes[2], *e.g.,* consider how the diverse text generation capability of a GPT is formed.

---

[1] Throughout this paper, generative modeling is used in its broad sense; for example, classification may be viewed as the generative modeling of a label conditioned on its feature.

[2] A misunderstanding is that $p_{\boldsymbol{\theta}}(y|\boldsymbol{x})$ has to be uni-model under the classification setup with feature $\boldsymbol{x}$ and label $y$. Note a multi-mode model can have a uni-model practical instantiation.

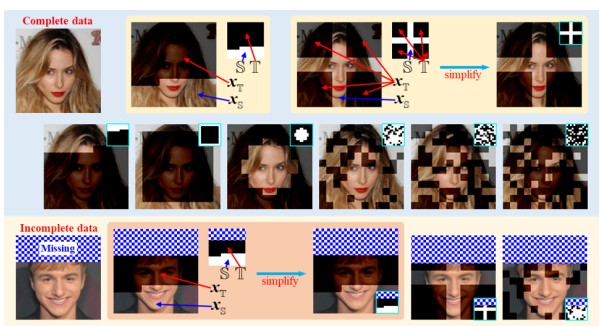
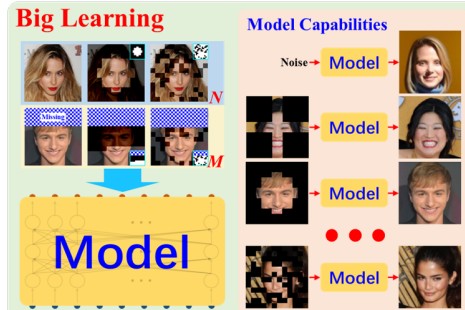

**(a)** A data can be exploited from many perspectives.  **(b)** The (uni-modal) big learning.

Figure 1: Big picture of the big learning, exampled by its uni-modal case. (a) When given a complete/incomplete data sample $\boldsymbol{x} \sim q(\boldsymbol{x})$, one simultaneously receives multiple joint, conditional, and marginal samples from $q(\boldsymbol{x}_{\mathbb{T}}|\boldsymbol{x}_{\mathbb{S}}), \forall(\mathbb{S}, \mathbb{T})$. (b) The big learning comprehensively exploits those samples to deliver versatile data capabilities with one model. See Appendix Fig. 6 for details.

2. **The GAN loss.** GANs are known for synthesizing highly realistic images with multiple modes [23; 24; 26]. A standard GAN consists of a generator $G_{\boldsymbol{\theta}}$ and a discriminator $D_{\boldsymbol{\phi}}$, both of which are trained in an adversarial manner via

$$\min_{\boldsymbol{\theta}} \max_{\boldsymbol{\phi}} \mathbb{E}_{q(\boldsymbol{x})} \log D_{\boldsymbol{\phi}}(\boldsymbol{x}) + \mathbb{E}_{p_{\boldsymbol{\theta}}(\boldsymbol{x})} \log(1 - D_{\boldsymbol{\phi}}(\boldsymbol{x})), \qquad (2)$$

where $q(\boldsymbol{x})$ is the underlying data distribution and $p_{\boldsymbol{\theta}}(\boldsymbol{x})$ is the generated distribution with the generative process $\boldsymbol{x} = G_{\boldsymbol{\theta}}(\boldsymbol{z}), \boldsymbol{z} \sim p(\boldsymbol{z})$. $p(\boldsymbol{z})$ is an easy-to-sample distribution, like a normal distribution. With optimal $D_{\boldsymbol{\phi}^*}$, Eq. (2) minimizes the Jensen-Shannon (JS) divergence $\text{JS}[q(\boldsymbol{x})\|p_{\boldsymbol{\theta}}(\boldsymbol{x})]$ [15].

To demonstrate the flexibilities of the big learning, we instantiate it within both maximum likelihood and adversarial learning territories (with the multi-mode objectives in (1) and (2), respectively) in Section 3.2, where Transformers/ViTs are employed to construct its universal foundation model.

## 3  BIG LEARNING

For better introduction of the big learning, we first present its main idea in simplified unsupervised/uni-modal settings, where a data sample $\boldsymbol{X} = (\boldsymbol{x})$ contains only a feature $\boldsymbol{x} \in \mathbb{R}^{L \times D}$ with length $L$ and dimension $D$. For example, $\boldsymbol{x}$ may represent $(i)$ a sentence consisting of $L$ words, each of which is encoded as a $D$-dimensional one-hot vector, or $(ii)$ an image patchified as $L$ image patches, each of which has $D$ pixels. Then, we generalize the scope of the big learning to the general settings, where a data sample $\boldsymbol{X} = (\boldsymbol{y}, \boldsymbol{x})$ contains both feature $\boldsymbol{x}$ and its paired supervision $\boldsymbol{y} \in \mathbb{R}^{L^{\boldsymbol{y}} \times D^{\boldsymbol{y}}}$ (*e.g.,* when $L^{\boldsymbol{y}} = D^{\boldsymbol{y}} = 1, y \in \{1, \cdots, C\}$ may represent a label). In both settings, the big learning naturally handles "incomplete data," which are defined as either $\boldsymbol{x}$ missing values along the $L$-dimension or $\boldsymbol{y}$ missing values along the $L^{\boldsymbol{y}}$-dimension.

### 3.1  UNSUPERVISED/UNI-MODAL BIG LEARNING

Given complete data samples $\boldsymbol{x} \in \mathbb{R}^{L \times D}$ drawn from the underlying data distribution $q(\boldsymbol{x})$, the mainstream machine learning paradigms concentrate on *joint matching, i.e.,* to construct a model $p_{\boldsymbol{\theta}}(\boldsymbol{x})$ in the joint domain (or $p_{\boldsymbol{\theta}}(\boldsymbol{x}_{\mathbb{L}})$ with $\mathbb{L} = \{1, \cdots, L\}$) to match $q(\boldsymbol{x})$, or *informally* $p_{\boldsymbol{\theta}}(\boldsymbol{x}) \longrightarrow q(\boldsymbol{x})$. Popular joint-matching learning paradigms include GANs [5; 23], Expectation-Maximization (EM) [11], VAEs [27; 10], Flows [13; 28], diffusion models [21; 43], *etc.*

However, joint matching can not take advantage of incomplete data (*e.g., $\boldsymbol{x}$* missing values along the $L$-dimension), which frequently arise in practical applications. Moreover, it may also fail to comprehensively exploit the information from a complete data sample, because diverse conditional/marginal samples (already given within that joint sample) are not explicitly utilized. In fact, based on the analyses in the Introduction, foundation models succeed in part from explicitly utilizing diverse conditional/marginal samples.

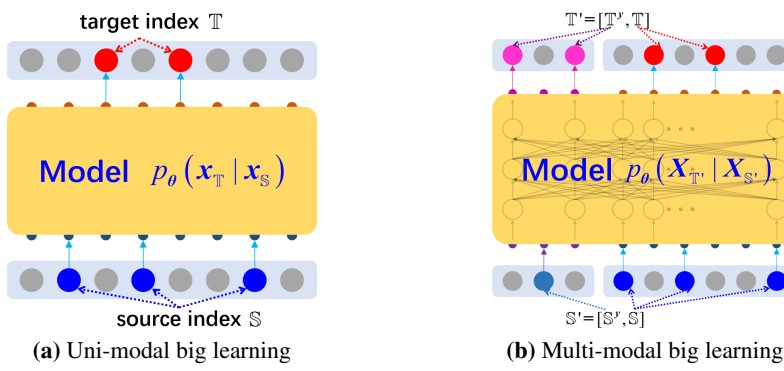

**(a)** Uni-modal big learning        **(b)** Multi-modal big learning

Figure 2: Demonstration of the network architectures.

To comprehensively exploit the data information within both complete and incomplete samples, we propose the following unsupervised/uni-modal big learning that leverages a universal foundation model to simultaneously model many/all joint, conditional, and marginal data distributions[3], manifested as *"big" learning with massive matching tasks*.

**Definition 1** (Unsupervised/Uni-modal big learning). *With the unsupervised/uni-modal setup, where a data sample $\boldsymbol{X} = (\boldsymbol{x})$ contains only a feature $\boldsymbol{x} \in \mathbb{R}^{L \times D}$ with length $L$ and dimension $D$, the length index set $\mathbb{L} = \{1, \cdots, L\}$, and any two non-overlapping subsets of $\mathbb{S} \subset \mathbb{L}$ and $\mathbb{T} \subseteq \mathbb{L}, \mathbb{T} \neq \emptyset$, the unsupervised/uni-modal big learning leverages a universal foundation model $p_{\boldsymbol{\theta}}(\boldsymbol{x}_{\mathbb{T}}|\boldsymbol{x}_{\mathbb{S}})$ (see Fig. 2a) to model many/all joint, conditional, and marginal data distributions[4] simultaneously,* i.e.,*

$$p_{\boldsymbol{\theta}}(\boldsymbol{x}_{\mathbb{T}}|\boldsymbol{x}_{\mathbb{S}}) \longrightarrow q(\boldsymbol{x}_{\mathbb{T}}|\boldsymbol{x}_{\mathbb{S}}), \forall (\mathbb{S}, \mathbb{T}) \in \boldsymbol{\Omega} \tag{3}$$

*where the arrow indicates utilizing its left-hand side to match its right-hand side. The actual objective measuring the distance/divergence (or encouraging the matching) between both sides of the arrow should be selected base on the application. $\boldsymbol{\Omega}$ is a user-defined set that contains the $(\mathbb{S}, \mathbb{T})$ pairs of interest. With different $(\mathbb{S}, \mathbb{T})$ pairs, $q(\boldsymbol{x}_{\mathbb{T}}|\boldsymbol{x}_{\mathbb{S}})$ may represent a joint/marginal/conditional data distribution, whose samples are readily available from the training data.*

*Remark* 1. In Theorem 1, $\mathbb{S} \cup \mathbb{T}$ need not be $\mathbb{L}$, meaning that incomplete data are naturally utilized.

*Remark* 2. Because input $\boldsymbol{x}_{\mathbb{S}}$ and output $\boldsymbol{x}_{\mathbb{T}}$ may have different dimensionalities for different $(\mathbb{S}, \mathbb{T})$ pairs, one may prefer constructing the universal $p_{\boldsymbol{\theta}}(\boldsymbol{x}_{\mathbb{T}}|\boldsymbol{x}_{\mathbb{S}})$ of (3) with a Transformer/ViT.

*Remark* 3. Possible choices for the objective associated with the arrow in (3) include the cross-entropy loss, the GAN loss, energy-based models [30], *etc*. Often one prefers employing the same objective for various $(\mathbb{S}, \mathbb{T})$ pairs.

*Remark* 4. Considering practical situations, one may alternatively or additionally do big learning in transformed domains, *e.g.*, via $p_{\boldsymbol{\theta}}(\hat{\boldsymbol{x}}_{\mathbb{T}}|\hat{\boldsymbol{x}}_{\mathbb{S}})$ with $\hat{\boldsymbol{x}} = g(\boldsymbol{x})$ or $p_{\boldsymbol{\theta}}(h(\boldsymbol{x}_{\mathbb{T}})|k(\boldsymbol{x}_{\mathbb{S}}))$ [20; 50], where $g(\cdot)$, $h(\cdot)$, and $k(\cdot)$ are domain-knowledge-inspired transformations.

## 3.2 Implementations of Unsupervised/Uni-modal Big Learning

We demonstrate unsupervised/uni-modal big learning with two example implementations, one of which is in the adversarial-learning territory with continuous observations, while the other is in the maximum-likelihood-learning territory with discrete observations.

### 3.2.1 Adversarial Learning for Foundation Models

Below we leverage the unsupervised/uni-modal big learning principle in Definition 1 to upgrade the standard GAN [15] into its big-learning variant termed the BigLearn-GAN, which is a novel adversarially-trained foundation model.

Given continuous observations $\boldsymbol{x} \in \mathbb{R}^{L \times D}$ (*e.g.*, $\boldsymbol{x}$ denoting an image patchified as $L$ patches), we design the universal model $p_{\boldsymbol{\theta}}(\boldsymbol{x}_{\mathbb{T}}|\boldsymbol{x}_{\mathbb{S}})$ based on ViT to model the *generative processes* of the output

---

[3]The incomplete data are readily utilized in the corresponding conditional/marginal tasks.

[4]To naively collect all the capabilities, one need construct at least $N_{\text{all}} = \sum_{i=0}^{L-1} C_L^i (\sum_{k=1}^{L-i} C_{L-i}^k)$ models, which is clearly prohibitive. See Appendix A for details.

Table 1: Big learning and its special cases. In general, $\boldsymbol{X} = (\boldsymbol{y}, \boldsymbol{x})$, $\boldsymbol{x} \in \mathbb{R}^{L \times D}$, $\boldsymbol{y} \in \mathbb{R}^{L^{\boldsymbol{y}} \times D^{\boldsymbol{y}}}$, $\mathbb{L}' = [\mathbb{L}^{\boldsymbol{y}}, \mathbb{L}]$, $\mathbb{S}' = [\mathbb{S}^{\boldsymbol{y}}, \mathbb{S}]$, and $\mathbb{T}' = [\mathbb{T}^{\boldsymbol{y}}, \mathbb{T}]$; with $\boldsymbol{X} = (\boldsymbol{x})$ and $\mathbb{L}^{\boldsymbol{y}} = \mathbb{S}^{\boldsymbol{y}} = \mathbb{T}^{\boldsymbol{y}} = \emptyset$, unsupervised big learning is recovered. When $y \in \{1, \cdots, C\}^{1 \times 1}$, it may represent a label. We ignore the implementation details and only focus on the core idea for demonstration.

| Big Learning | $p_{\boldsymbol{\theta}}(\boldsymbol{X}_{\mathbb{T}'}|\boldsymbol{X}_{\mathbb{S}'}) \longrightarrow q(\boldsymbol{X}_{\mathbb{T}'}|\boldsymbol{X}_{\mathbb{S}'}), \forall (\mathbb{S}', \mathbb{T}')$ | $\mathbb{S}' \subset \mathbb{L}', \mathbb{T}' \subseteq \mathbb{L}', \mathbb{T}' \neq \emptyset$, and $\mathbb{S}' \cap \mathbb{T}' = \emptyset$ |
|---|---|---|
| ↓Special Case | ↓Training Objective | ↓Constraints |
| Masked LM [44] | $\mathbb{E}_{q(\mathbb{S}, \mathbb{T})} \mathrm{KL}[q(\boldsymbol{x}_{\mathbb{T}}|\boldsymbol{x}_{\mathbb{S}})||p_{\boldsymbol{\theta}}(\boldsymbol{x}_{\mathbb{T}}|\boldsymbol{x}_{\mathbb{S}})]$ | $q(\mathbb{S}, \mathbb{T}) = \mathcal{U}\{(\mathbb{S}, \mathbb{T}) : \mathbb{S} \text{ is a } 85\% \text{ random subset of } \mathbb{L}, \text{ and } \mathbb{T} = \mathbb{L} \backslash \mathbb{S}\}$ $p_{\boldsymbol{\theta}}(\boldsymbol{x}_{\mathbb{T}}|\boldsymbol{x}_{\mathbb{S}}) = \prod_{t \in \mathbb{T}} \text{Categorical}(x_t|\boldsymbol{p}_{\boldsymbol{\theta}}(\boldsymbol{x}_{\mathbb{S}}))$ |
| Causal/Auto-regressive LM [6; 39; 35; 37] | $\sum_{(\mathbb{S}, \mathbb{T}) \in \Xi} \mathrm{KL}[q(\boldsymbol{x}_{\mathbb{T}}|\boldsymbol{x}_{\mathbb{S}})||p_{\boldsymbol{\theta}}(\boldsymbol{x}_{\mathbb{T}}|\boldsymbol{x}_{\mathbb{S}})]$ | $\Xi = \{(\emptyset, 1), (\{1\}, 2), (\{1, 2\}, 3), \cdots\}$ $p_{\boldsymbol{\theta}}(\boldsymbol{x}_{\mathbb{T}}|\boldsymbol{x}_{\mathbb{S}}) = \text{Categorical}(\boldsymbol{x}_{\mathbb{T}}|\boldsymbol{p}_{\boldsymbol{\theta}}(\boldsymbol{x}_{\mathbb{S}}))$ |
| Permutation LM [52] | $\mathbb{E}_{q(\mathbb{S}, \mathbb{T})} \sum_{(\tilde{\mathbb{S}}, \tilde{\mathbb{T}}) \in \Xi_{\mathbb{S}, \mathbb{T}}} \mathrm{KL}[q(\boldsymbol{x}_{\tilde{\mathbb{T}}}|\boldsymbol{x}_{\tilde{\mathbb{S}}})||p_{\boldsymbol{\theta}}(\boldsymbol{x}_{\tilde{\mathbb{T}}}|\boldsymbol{x}_{\tilde{\mathbb{S}}})]$ | $q(\mathbb{S}, \mathbb{T}) = \mathcal{U}\{(\mathbb{S}, \mathbb{T}) : \mathbb{S} \text{ is a } 85\% \text{ random subset of } \mathbb{L}, \text{ and }$ $\mathbb{T} = \{t_1, t_2, \cdots\} \text{ is a random permutation of } \mathbb{L} \backslash \mathbb{S}\}$ $\Xi_{\mathbb{S}, \mathbb{T}} = \{(\mathbb{S}, t_1), (\{\mathbb{S}, t_1\}, t_2), (\{\mathbb{S}, t_1, t_2\}, t_3), \cdots\}$ $p_{\boldsymbol{\theta}}(\boldsymbol{x}_{\tilde{\mathbb{T}}}|\boldsymbol{x}_{\tilde{\mathbb{S}}}) = \text{Categorical}(\boldsymbol{x}_{\tilde{\mathbb{T}}}|\boldsymbol{p}_{\boldsymbol{\theta}}(\boldsymbol{x}_{\tilde{\mathbb{S}}}))$ |
| MAE [20] MaskFeat [50] | $\mathbb{E}_{q(\mathbb{S}, \mathbb{T})} \mathrm{KL}[q(h(\boldsymbol{x}_{\mathbb{T}})|\boldsymbol{x}_{\mathbb{S}})||p_{\boldsymbol{\theta}}(h(\boldsymbol{x}_{\mathbb{T}})|\boldsymbol{x}_{\mathbb{S}})]$ | $q(\mathbb{S}, \mathbb{T}) = \mathcal{U}\{(\mathbb{S}, \mathbb{T}) : \mathbb{S} \text{ is a } 25\% \text{ random subset of } \mathbb{L}, \text{ and } \mathbb{T} = \mathbb{L} \backslash \mathbb{S}$ $p_{\boldsymbol{\theta}}(h(\boldsymbol{x}_{\mathbb{T}})|\boldsymbol{x}_{\mathbb{S}}) = \mathcal{N}(h(\boldsymbol{x}_{\mathbb{T}})|\boldsymbol{\mu}_{\boldsymbol{\theta}}(\boldsymbol{x}_{\mathbb{S}}), \mathbf{I})$ $h(\cdot) \text{ is a normalization/HOG transformation for MAE/MaskFeat}$ |
| Big Learning with (4) | $\mathbb{E}_{q(\mathbb{S}, \mathbb{T})} \mathrm{JS}[q(\boldsymbol{x}_{\mathbb{S} \cup \mathbb{T}})||p_{\boldsymbol{\theta}}(\boldsymbol{x}_{\mathbb{T}}|\boldsymbol{x}_{\mathbb{S}})q(\boldsymbol{x}_{\mathbb{S}})]$ | $q(\mathbb{S}, \mathbb{T}) = \mathcal{U}\{(\mathbb{S}, \mathbb{T}) : \mathbb{S} \text{ is a random subset of } \mathbb{L}, \text{ and } \mathbb{T} \text{ is a random}$ $\text{subset of } \mathbb{L} \backslash \mathbb{S}\}$ $p_{\boldsymbol{\theta}}(\boldsymbol{x}_{\mathbb{T}}|\boldsymbol{x}_{\mathbb{S}}) \text{ is a universal ViT-based GAN generator}$ |
| Big Learning with (6) | $\mathbb{E}_{q(\mathbb{S}, \mathbb{T})} \sum_{(\tilde{\mathbb{S}}, \tilde{\mathbb{T}}) \in \Xi_{\mathbb{S}, \mathbb{T}}} \mathrm{KL}[q(\boldsymbol{x}_{\tilde{\mathbb{T}}}|\boldsymbol{x}_{\tilde{\mathbb{S}}})||p_{\boldsymbol{\theta}}(\boldsymbol{x}_{\tilde{\mathbb{T}}}|\boldsymbol{x}_{\tilde{\mathbb{S}}})]$ | $q(\mathbb{S}, \mathbb{T}) = \mathcal{U}\{(\mathbb{S}, \mathbb{T}) : \mathbb{S} \text{ is a random subset of } \mathbb{L}, \text{ and }$ $\mathbb{T} = \{t_1, t_2, \cdots\} \text{ is a random permuted subset of } \mathbb{L} \backslash \mathbb{S}\}$ $\Xi_{\mathbb{S}, \mathbb{T}} = \{(\mathbb{S}, t_1), (\{\mathbb{S}, t_1\}, t_2), (\{\mathbb{S}, t_1, t_2\}, t_3), \cdots\}$ $p_{\boldsymbol{\theta}}(\boldsymbol{x}_{\tilde{\mathbb{T}}}|\boldsymbol{x}_{\tilde{\mathbb{S}}}) = \text{Categorical}(\boldsymbol{x}_{\tilde{\mathbb{T}}}|\boldsymbol{p}_{\boldsymbol{\theta}}(\boldsymbol{x}_{\tilde{\mathbb{S}}}))$ |
| Supervised Classification | *e.g.*, $\mathrm{KL}[q(\boldsymbol{y}|\boldsymbol{x})||p_{\boldsymbol{\theta}}(\boldsymbol{y}|\boldsymbol{x})]$ | $\mathbb{S}' = [\emptyset, \mathbb{L}], \mathbb{T}' = [\mathbb{L}^{\boldsymbol{y}}, \emptyset], p_{\boldsymbol{\theta}}(\boldsymbol{y}|\boldsymbol{x}) \text{ is } e.g., \text{ a classifier}$ |
| Joint Generation | *e.g.*, $\mathrm{JS}[q(\boldsymbol{x})||p_{\boldsymbol{\theta}}(\boldsymbol{x})]$ | $\mathbb{S}' = [\emptyset, \emptyset], \mathbb{T}' = [\emptyset, \mathbb{L}], p_{\boldsymbol{\theta}}(\boldsymbol{x}) \text{ may be a generator}$ |
| Conditioned Generation | *e.g.*, $\mathrm{KL}[q(\boldsymbol{x}|\boldsymbol{y})||p_{\boldsymbol{\theta}}(\boldsymbol{x}|\boldsymbol{y})]$ | $\mathbb{S}' = [\mathbb{L}^{\boldsymbol{y}}, \emptyset], \mathbb{T}' = [\emptyset, \mathbb{L}], p_{\boldsymbol{\theta}}(\boldsymbol{x}|\boldsymbol{y}) : \text{a conditional flow}$ |

$\boldsymbol{x}_{\mathbb{T}}$ given the input $\boldsymbol{x}_{\mathbb{S}}$ for *all* $(\mathbb{S}, \mathbb{T})$ pairs (see Appendix C for the detailed architecture). Note when $\mathbb{T} = \mathbb{L}$ and $\mathbb{S} = \emptyset$, $p_{\boldsymbol{\theta}}(\boldsymbol{x}_{\mathbb{T}}|\boldsymbol{x}_{\mathbb{S}})$ reduces to the commonly-used joint generator. The standard GAN loss is employed as the objective that is associated with the arrow in (3).

Following (3), one may naively specify the big-learning objective as

$$\min_{\boldsymbol{\theta}} \max_{\boldsymbol{\phi}} \mathbb{E}_{q(\mathbb{S}, \mathbb{T})} \big[ \mathbb{E}_{q(\boldsymbol{x}_{\mathbb{S} \cup \mathbb{T}})} \log \sigma[f_{\boldsymbol{\phi}}(\boldsymbol{x}; \mathbb{S}, \mathbb{T})] + \mathbb{E}_{p_{\boldsymbol{\theta}}(\boldsymbol{x}_{\mathbb{T}}|\boldsymbol{x}_{\mathbb{S}})q(\boldsymbol{x}_{\mathbb{S}})} \log \sigma[-f_{\boldsymbol{\phi}}(\boldsymbol{x}; \mathbb{S}, \mathbb{T})] \big], \quad (4)$$

which, in the ideal situation, performs $\min_{\boldsymbol{\theta}} \max_{\boldsymbol{\phi}} \mathbb{E}_{q(\mathbb{S}, \mathbb{T})} \mathrm{JS}[q(\boldsymbol{x}_{\mathbb{S} \cup \mathbb{T}})||p_{\boldsymbol{\theta}}(\boldsymbol{x}_{\mathbb{T}}|\boldsymbol{x}_{\mathbb{S}})q(\boldsymbol{x}_{\mathbb{S}})]$, encouraging the matchings between $p_{\boldsymbol{\theta}}(\boldsymbol{x}_{\mathbb{T}}|\boldsymbol{x}_{\mathbb{S}})$ and $q(\boldsymbol{x}_{\mathbb{T}}|\boldsymbol{x}_{\mathbb{S}})$ for *many/all* $(\mathbb{S}, \mathbb{T})$ pairs. $q(\mathbb{S}, \mathbb{T})$ denotes the sampling process of $(\mathbb{S}, \mathbb{T})$ (see Appendix D) and it implicitly defines the weighting among joint, marginal, and conditional matchings. The optimal $f_{\boldsymbol{\phi}^*}(\boldsymbol{x}; \mathbb{S}, \mathbb{T}) = \log \frac{q(\boldsymbol{x}_{\mathbb{S} \cup \mathbb{T}})}{p_{\boldsymbol{\theta}}(\boldsymbol{x}_{\mathbb{T}}|\boldsymbol{x}_{\mathbb{S}})q(\boldsymbol{x}_{\mathbb{S}})} = \log \frac{q(\boldsymbol{x}_{\mathbb{T}}|\boldsymbol{x}_{\mathbb{S}})}{p_{\boldsymbol{\theta}}(\boldsymbol{x}_{\mathbb{T}}|\boldsymbol{x}_{\mathbb{S}})}$.

Noticing that the universal $p_{\boldsymbol{\theta}}(\boldsymbol{x}_{\mathbb{T}}|\boldsymbol{x}_{\mathbb{S}})$ possesses versatile data sampling capabilities, we take a step further and propose to *explicitly* enhance those sampling capabilities during learning, mimicking the core idea of the big learning principle. Specifically, we leverage those sampling capabilities to introduce additional learning tasks, by considering that any two model distributions $p_{\boldsymbol{\theta}}(\boldsymbol{x}_{\mathbb{T}_1}|\boldsymbol{x}_{\mathbb{S}_1})q(\boldsymbol{x}_{\mathbb{S}_1})$ and $p_{\boldsymbol{\theta}}(\boldsymbol{x}_{\mathbb{T}_2}|\boldsymbol{x}_{\mathbb{S}_2})q(\boldsymbol{x}_{\mathbb{S}_2})$ with $\mathbb{S}^1 \cup \mathbb{T}^1 = \mathbb{S}^2 \cup \mathbb{T}^2$ should be close to each other, because they share the same ultimate goal of matching $q(\boldsymbol{x}_{\mathbb{S}^1 \cup \mathbb{T}^1})$.

Accordingly, we enable additional "communications" among any two functionalities of the universal model $p_{\boldsymbol{\theta}}(\boldsymbol{x}_{\mathbb{T}}|\boldsymbol{x}_{\mathbb{S}})$ and present the additional big learning objective as

$$\min_{\boldsymbol{\theta}} \max_{\boldsymbol{\phi}} \mathbb{E}_{q(\mathbb{S}^1, \mathbb{T}^1)q(\mathbb{S}^2, \mathbb{T}^2)} \begin{bmatrix} \mathbb{E}_{p_{\boldsymbol{\theta}}(\boldsymbol{x}_{\mathbb{T}^1}|\boldsymbol{x}_{\mathbb{S}^1})q(\boldsymbol{x}_{\mathbb{S}^1})} \log \sigma[f_{\boldsymbol{\phi}}(\boldsymbol{x}; \mathbb{S}^2, \mathbb{T}^2) - f_{\boldsymbol{\phi}}(\boldsymbol{x}; \mathbb{S}^1, \mathbb{T}^1)] \\ + \mathbb{E}_{p_{\boldsymbol{\theta}}(\boldsymbol{x}_{\mathbb{T}^2}|\boldsymbol{x}_{\mathbb{S}^2})q(\boldsymbol{x}_{\mathbb{S}^2})} \log \sigma[f_{\boldsymbol{\phi}}(\boldsymbol{x}; \mathbb{S}^1, \mathbb{T}^1) - f_{\boldsymbol{\phi}}(\boldsymbol{x}; \mathbb{S}^2, \mathbb{T}^2)] \end{bmatrix}, \quad (5)$$

where the "communication" discriminator can be implicitly constructed with the same neural network $f_{\boldsymbol{\phi}}(\boldsymbol{x}; \mathbb{S}, \mathbb{T})$ from (4). Proofs are given in Appendix B.

Combining (4) and (5) yields the tailored big learning objective for the BigLearn-GAN, which is the first principled adversarial pretraining strategy for foundation models, to our knowledge.

### 3.2.2 MAXIMUM-LIKELIHOOD IMPLEMENTATION

Consider applications with discrete observations $\boldsymbol{x} \in \mathbb{Z}^{L \times 1}$; for example, $\boldsymbol{x}$ denotes a sentence with $L$ words or an image that is vector-quantified into a sequence of indexes [39]. Eq. (3) of Definition

1 motivate us to model the distribution $p_{\boldsymbol{\theta}}(\boldsymbol{x}_{\mathbb{T}}|\boldsymbol{x}_{\mathbb{S}})$ of multiple output words $\boldsymbol{x}_{\mathbb{T}}$ conditioned on input words $\boldsymbol{x}_{\mathbb{S}}$, which is challenging considering the correlations among $\boldsymbol{x}_{\mathbb{T}}$-words.

- One brute-force solution is to ignore those correlations, which in turn degrades the unsupervised/uni-modal big learning in (3) into the Masked LM [44] with multiple masking ratios.

- An alternative solution is to auto-regressively model those correlations, which in turn degrades the unsupervised/uni-modal big learning into the permutation LM [52] that considers various prediction orderings.[5]

We demonstrate with the letter solution. With a Transformer-based universal model $p_{\boldsymbol{\theta}}(\boldsymbol{x}_{\bar{\mathbb{T}}}|\boldsymbol{x}_{\bar{\mathbb{S}}})$ modeling the generative process of *one* output word $\boldsymbol{x}_{\bar{\mathbb{T}}}$ given input words $\boldsymbol{x}_{\bar{\mathbb{S}}}$ for *any* $(\bar{\mathbb{S}}, \bar{\mathbb{T}})$ pair, the tailored big learning objective may be defined as

$$\max_{\boldsymbol{\theta}} \mathbb{E}_{q(\mathbb{S},\mathbb{T})} \sum_{(\bar{\mathbb{S}},\bar{\mathbb{T}}) \in \Xi_{\mathbb{S},\mathbb{T}}} \mathbb{E}_{q(\boldsymbol{x}_{\bar{\mathbb{T}}}|\boldsymbol{x}_{\bar{\mathbb{S}}})} \log p_{\boldsymbol{\theta}}(\boldsymbol{x}_{\bar{\mathbb{T}}}|\boldsymbol{x}_{\bar{\mathbb{S}}}), \tag{6}$$

where $q(\mathbb{S}, \mathbb{T})$ denotes the sampling process of $(\mathbb{S}, \mathbb{T})$ with random permutations, $\mathbb{T} = \{t_1, t_2, \cdots\}$, $\Xi_{\mathbb{S},\mathbb{T}} = \{(\mathbb{S}, t_1), (\{\mathbb{S}, t_1\}, t_2), (\{\mathbb{S}, t_1, t_2\}, t_3), \cdots\}$, often $p_{\boldsymbol{\theta}}(\boldsymbol{x}_{\bar{\mathbb{T}}}|\boldsymbol{x}_{\bar{\mathbb{S}}}) = \text{Categorical}(\boldsymbol{x}_{\bar{\mathbb{T}}}|\boldsymbol{p}_{\boldsymbol{\theta}}(\boldsymbol{x}_{\bar{\mathbb{S}}}))$ is modeled as a categorical distribution with probabilities $\boldsymbol{p}_{\boldsymbol{\theta}}(\boldsymbol{x}_{\bar{\mathbb{S}}})$, and $\boldsymbol{x}_{\bar{\mathbb{T}}}$ always contain one word.

After unsupervised/uni-modal big learning, the universal $p_{\boldsymbol{\theta}}(\boldsymbol{x}_{\bar{\mathbb{T}}}|\boldsymbol{x}_{\bar{\mathbb{S}}})$ may possess versatile generation and data completion capabilities *w.r.t. any* predicting order.

### 3.3 GENERAL/MULTI-MODAL BIG LEARNING

Thanks to the modeling flexibility of the unsupervised/uni-modal big learning, it's convenient to generalize it into the general/multi-modal big learning, where $\boldsymbol{X} = (\boldsymbol{y}, \boldsymbol{x})$ contains an additional supervision $\boldsymbol{y} \in \mathbb{R}^{L^{\boldsymbol{y}} \times D^{\boldsymbol{y}}}$. The key idea is to interpret paired multi-modal data as a "larger" sample.

**Definition 2** (General/Multi-modal big learning). *With the general/multi-modal setup, where a data sample $\boldsymbol{X} = (\boldsymbol{y}, \boldsymbol{x})$[6] contains both feature $\boldsymbol{x} \in \mathbb{R}^{L \times D}$ and its paired supervision $\boldsymbol{y} \in \mathbb{R}^{L^{\boldsymbol{y}} \times D^{\boldsymbol{y}}}$ with the $\boldsymbol{X}$-length index set $\mathbb{L}' = [\mathbb{L}^{\boldsymbol{y}}, \mathbb{L}]$, its any two non-overlapping input/output index subsets $\mathbb{S}' = [\mathbb{S}^{\boldsymbol{y}}, \mathbb{S}]$ and $\mathbb{T}' = [\mathbb{T}^{\boldsymbol{y}}, \mathbb{T}]$ with $\mathbb{S}' \subset \mathbb{L}'$, $\mathbb{T}' \subseteq \mathbb{L}'$, and $\mathbb{T}' \neq \emptyset$, the general/multi-modal big learning leverages a universal foundation model $p_{\boldsymbol{\theta}}(\boldsymbol{X}_{\mathbb{T}'}|\boldsymbol{X}_{\mathbb{S}'})$ (see Fig. 2b) to model many/all joint, conditional, and marginal $\boldsymbol{X}$-data distributions simultaneously,* i.e.,

$$p_{\boldsymbol{\theta}}(\boldsymbol{X}_{\mathbb{T}'}|\boldsymbol{X}_{\mathbb{S}'}) \longrightarrow q(\boldsymbol{X}_{\mathbb{T}'}|\boldsymbol{X}_{\mathbb{S}'}), \forall (\mathbb{S}', \mathbb{T}') \in \boldsymbol{\Omega}' \tag{7}$$

*where $\boldsymbol{\Omega}'$ is a user-defined set containing all $(\mathbb{S}', \mathbb{T}')$ pairs or a portion of them. $q(\boldsymbol{X}) \triangleq q(\boldsymbol{y}, \boldsymbol{x})$ is the underlying complete data distribution. For any $(\mathbb{S}', \mathbb{T}')$, $q(\boldsymbol{X}_{\mathbb{T}'}|\boldsymbol{X}_{\mathbb{S}'})$ is the corresponding joint/conditional/marginal $\boldsymbol{X}$-data distribution, whose samples are readily available from the training dataset.*

*Remark* 5. For situations where $\boldsymbol{X} = (\boldsymbol{y}, \boldsymbol{x})$ has the same data type (*e.g.,* both $\boldsymbol{y}$ and $\boldsymbol{x}$ denote *continuous* patchified images), the general/multi-modal big learning works the same as its unsupervised/uni-modal simplification. However, for challenging situations where each modality has a different data type, *e.g.,* where $\boldsymbol{y}$ denotes a sequence of *discrete* text words but $\boldsymbol{x}$ are a sequence of *continuous* image-patches [17; 32; 39; 40; 1], one may resort to the following two techniques to enjoy the general/multi-modal big learning.

1. **To transform one data type into the other type for alignment**, *e.g.,* one can vector-quantize the *continuous* $\boldsymbol{x}$ into a sequence of *discrete* tokens [39], followed by resorting to (6).

2. **To recursively reuse $p_{\boldsymbol{\theta}}(\boldsymbol{X}_{\mathbb{T}'}|\boldsymbol{X}_{\mathbb{S}'})$ to isolate each type**, *i.e.,* one can unfold the learning via

$$p_{\boldsymbol{\theta}}(\boldsymbol{X}_{\mathbb{T}'}|\boldsymbol{X}_{\mathbb{S}'}) = p_{\boldsymbol{\theta}}(\boldsymbol{y}_{\mathbb{T}^{\boldsymbol{y}}}|\boldsymbol{x}_{\mathbb{T}}, \boldsymbol{X}_{\mathbb{S}'}) p_{\boldsymbol{\theta}}(\boldsymbol{x}_{\mathbb{T}}|\boldsymbol{X}_{\mathbb{S}'}) = p_{\boldsymbol{\theta}}(\boldsymbol{X}_{\mathbb{T}^{\boldsymbol{y}}}|\boldsymbol{X}_{\mathbb{T} \cup \mathbb{S}'}) p_{\boldsymbol{\theta}}(\boldsymbol{X}_{\mathbb{T}}|\boldsymbol{X}_{\mathbb{S}'}), \tag{8}$$

where $\boldsymbol{X}_{\mathbb{T}^{\boldsymbol{y}}}/\boldsymbol{X}_{\mathbb{T}}$ has one unique data type after unfolding. One can then resort to big learning both $p_{\boldsymbol{\theta}}(\boldsymbol{X}_{\mathbb{T}^{\boldsymbol{y}}}|\boldsymbol{X}_{\mathbb{T} \cup \mathbb{S}'}) \longrightarrow q(\boldsymbol{X}_{\mathbb{T}^{\boldsymbol{y}}}|\boldsymbol{X}_{\mathbb{T} \cup \mathbb{S}'})$ and $p_{\boldsymbol{\theta}}(\boldsymbol{X}_{\mathbb{T}}|\boldsymbol{X}_{\mathbb{S}'}) \longrightarrow q(\boldsymbol{X}_{\mathbb{T}}|\boldsymbol{X}_{\mathbb{S}'})$ for training.

---

[5]The GAN implementation with (4) and (5) need not consider the ordering of $\mathbb{T}$ thanks to its (conditionally) joint matching nature.

[6]We present with two modalities for simplicity; the presented big learning can be readily generalized to situations with multiple paired modalities.

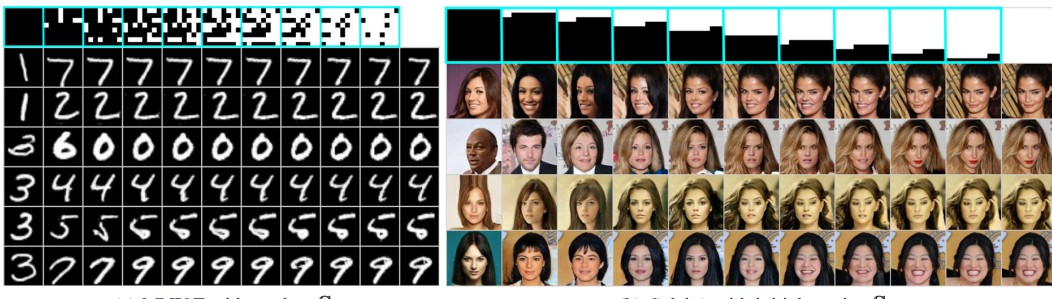

**(a)** MNIST with random $\mathbb{S}$                    **(b)** CelebA with initial-portion $\mathbb{S}$

Figure 3: Versatile data generation/completion capabilities from big learning. The first row with light-blue boxes shows different $\mathbb{S}$s, with an increasing $\mathbb{S}$-ratio from left to right. The rightmost column gives the real image.

### 3.4 DISCUSSIONS ON THE BIG LEARNING

Without loss of generality, we focus on the simplified unsupervised/uni-modal settings for presentation and only employ the complicated general/multi-modal one if necessary.

**Can we share one universal foundation model $p_{\boldsymbol{\theta}}(\boldsymbol{x}_{\mathbb{T}}|\boldsymbol{x}_{\mathbb{S}})$ among all $(\mathbb{S}, \mathbb{T})$ pairs? Yes, and it's what we should do.** In the ideal situation, *all* conditional/marginal data distributions $q(\boldsymbol{x}_{\mathbb{T}}|\boldsymbol{x}_{\mathbb{S}})$ can be analytically derived from the (underlying) joint one $q(\boldsymbol{x})$, meaning that they all share the same set of underlying "parameters". Accordingly, their modelings are also expected to share parameters. Besides, sharing parameters also enables cross-regularization among joint, conditional, and marginal matchings, which likely encourages model parameters to approach that underlying "parameters."

**On big-learned model parameters and latent features.** Most foundation models, exhibiting extraordinary robustness, adaptability, and generalization, are trained with objectives that special cases of the big learning. Accordingly, we try to explain from the big learning perspective why they have such amazing characteristics.

- Firstly, by referring to (3) and (7), both the model parameters and its latent features are shared among many/all joint, conditional, and marginal matching tasks, all of which have the same consistent goal of modeling the intrinsic data information (*i.e.,* the aforementioned underlying "parameters") from diverse perspectives. Therefore, it's expected that big learning would encourage the model parameters or its latent features to approach the intrinsic information associated with those "parameters," which is manifested as those amazing characteristics.
- Secondly, the extraordinary data and task flexibilities of the big learning enable large-scale training with massive (complete/incomplete) data and diverse tasks (across potentially many domains). The significantly expanded training experiences (associated with both data and tasks) are expected to effectively reduce the training-test (or pretraining-finetuning) gap and therefore improve the robustness/generalization of big-learned foundation models.

**Big learning versus self-supervised contrastive learning.** Contrastive learning focuses on exploiting domain prior knowledge to learn generally applicable data representations for downstream tasks [19; 8; 16; 9]. From the perspective of prior exploitation, contrastive learning is orthogonal to the big learning that is mostly data-driven. One can of course consider leveraging the flexibility of big learning to combine it with contrastive learning to incorporate trustworthy domain priors.

### 4 EXPERIMENTS

The data/task flexibilities of the big learning significantly expand its scope of application, which, however, also brings tremendous challenges to the comprehensive evaluation of its properties.

Here we concentrate on demonstrating several exploration achievements, most of which are associated with the BigLearn-GAN developed in Section 3.2.1. Specifically, we first reveal that unsupervised/uni-modal big learning is indeed capable of delivering *all* joint, conditional, and marginal data capabilities via one universal foundation model trained on the MNIST/CelebA datasets (see Appendix D for experimental details). We then demonstrate the somewhat generalization capability of that big-learned foundation model with diverse abused out-of-domain challenges.

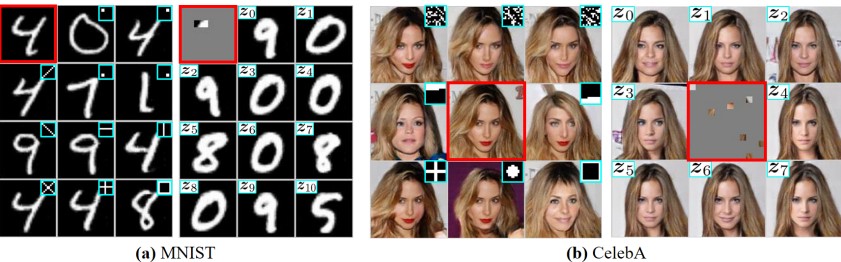

**(a)** MNIST    **(b)** CelebA

Figure 4: Versatile data completion capabilities from big learning *w.r.t.* various $\mathbb{S}$ (left) and noise $z$ (right). $\mathbb{S}$s are shown in upper-right light-blue boxes, while the red boxes show $x$ (left) and $x_{\mathbb{S}}$ (right), respectively.

Next, based on the maximum-likelihood implementation in (6), we show that big learning can naturally handle multi-modal data and its joint, conditional, and marginal data capabilities directly manifest as versatile functionalities like classification and generation. Finally, considering the quantitative evaluations of the big learning, we conduct experiments on the GLUE benchmark to reveal that big learning can serve as a superior fine-tuning strategy than the naive one.

*We highlight that, in addition to the BigLearn-GAN that leverages the big learning principle to upgrade the conventional GAN, we also provide in the supplementary materials similar research achievements on leveraging the big learning to upgrade the EM algorithm and the VAE, which justifies the effectiveness of the big learning with diverse experiments across different research domains.*

### 4.1 VERSATILE COMPLETION CAPABILITIES WITH ADAPTIVE GENERATION DIVERSITY

We first test the big-learned data generation/completion capabilities with different ratios $r_{\mathbb{S}}$ of $\mathbb{S}$ in $\mathbb{L}$. For a specific $r_{\mathbb{S}}$, we either randomly sample $r_{\mathbb{S}}L$ image patches or choose the first $r_{\mathbb{S}}$-portion to form the source $x_{\mathbb{S}}$, which is then input to the model $p_{\boldsymbol{\theta}}(x_{\mathbb{T}}|x_{\mathbb{S}})$ for image completion.

Fig. 3 shows the corresponding results. It's clear that the big-learned model masters many/all joint, conditional, and marginal data capabilities simultaneously. Besides, big learning also learns from the data an adaptive generation diversity conditioned on $x_{\mathbb{S}}$. Specifically, with increasing/decreasing $r_{\mathbb{S}}$ (*i.e.,* more/less source information), big learning delivers increasingly deterministic/diverse generations controlled by $x_{\mathbb{S}}$/random-noise, following our intuition (see Appendix G for more results).

We then test the big-learned capabilities with respect to various $\mathbb{S}$ and noise settings, with the results summarized in Fig. 4. On the one hand, given an image $x$ and a random noise $z$, big learning clearly delivers for various $\mathbb{S}$s diverse realistic generations on both MNIST (see the variations in class/stroke-thickness/shape/angle) and CelebA (see the varying identity/hair-style/make-up/expression). On the other hand, given a specific $x_{\mathbb{S}}$ with limited information, the big-learned model, when input different noises $z_i$, also generates realistic images with diversity.

The experimental results in Figs. 3 and 4 demonstrate that, by comprehensively exploiting the available information inherent in large-scale complete/incomplete data, big learning is capable of delivering versatile data generation/completion capabilities with learned adaptive generation diversity.

### 4.2 GENERALIZATION ON ABUSED ANOMALOUS OUT-OF-DOMAIN COMPLETION

We design abused completion tasks to test the generalization of the big learning. Specifically, we intentionally design $x_{\mathbb{S}}$ with ($i$) abused interventions to source patches (*e.g.,* random relocation and duplication, as shown in Fig. 5(a)); ($ii$) mixed-up patches from different data samples (see Fig. 5(b)); and ($iii$) unseen out-of-domain image patches, as shown in Fig. 5(c).

It's clear that big learning manages to handle these abused $x_{\mathbb{S}}$ with reasonable image completion; *e.g.,* see the realistic characters with overall consistent style and smooth strokes in Fig. 5(a), the harmoniously completed faces even with mismatched face frame and hair color in Fig. 5(b), and the relatively smooth out-of-domain completion in Fig. 5(c). These surprising results from abused anomalous out-of-domain completions (along with the great successes of existing foundation models) validate the generalization capability of the presented big learning.

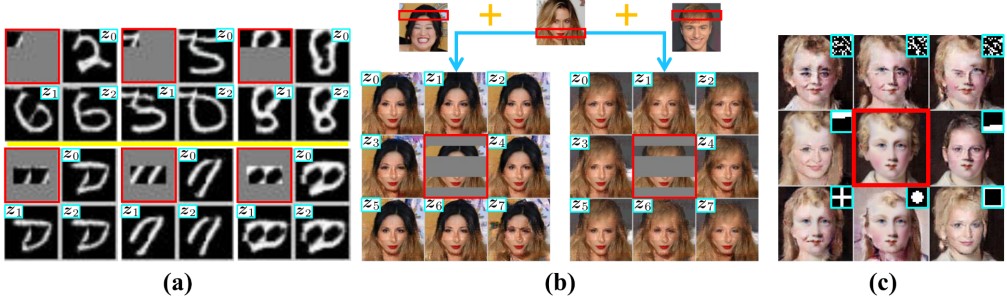

**(a)**          **(b)**          **(c)**

Figure 5: Abused anomalous completion for demonstrating the generalization of big learning. (a) $x_{\mathbb{S}}$ constructed with random center patches replaced in the upper-left corner (top) and duplicated and replaced in the center (bottom). A model big-learned on CelebA is used in (b)-(c). (b) $x_{\mathbb{S}}$ combining patches from different images. (c) Out-of-domain $x_{\mathbb{S}}$ from MetFaces [25].

Table 2: Big learning serves as a superior fine-tuning strategy. The best/median metrics are calculated among the combinations of the tested hyperparameters of Table 4.

| Task | Best Accuracy / F1 | | Median Accuracy / IQR | |
|---|---|---|---|---|
| | FT | big-learn | FT | big-learn |
| RTE | 71.84 | **75.09** | 66.06/2.34 | **70.75/1.44** |
| MRPC | 88.97/92.09 | **90.20/93.03** | 87.00/2.45 | **87.74/1.10** |
| SST-2 | 94.15 | **95.18** | 93.75/0.45 | **94.66/0.28** |

### 4.3 LEVERAGING BIG LEARNING TO UNIFY CLASSIFICATION AND GENERATION

We test the big learning in the general settings, where $X = (y, x)$ contains both image tokens $x \in \mathbb{Z}^{L \times 1}$ and a paired label $y \in \{1, \cdots, C\}^{1 \times 1}$. We conduct the experiment on MNIST and follow [3; 39] to first vector-quantize an image for its tokens $x$, followed by big learning based on (6). Details are given in Appendix E.

Given the big-learned universal model $p_{\theta}(X_{\mathbb{T}'} | X_{\mathbb{S}'})$, one can retrieve from it versatile data capabilities by specifying the corresponding $(\mathbb{S}', \mathbb{T}')$, such as joint generation (*i.e.,* $p_{\theta}(x)$; see Appendix Fig. 11(a) for the results), label-conditioned generation (*i.e.,* $p_{\theta}(x|y)$; see Fig. 11(b)), classification (*i.e.,* $p_{\theta}(y|x)$), random completion (*i.e.,* $p_{\theta}(x_{\mathbb{T}}|x_{\mathbb{S}})$), label-conditioned completion (*i.e.,* $p_{\theta}(x_{\mathbb{T}}|x_{\mathbb{S}}, y)$), *etc.* These simultaneously-delivered capabilities are likely valuable for counterfactual analysis and reasoning.

### 4.4 QUANTITATIVE EVALUATIONS ON THE GLUE BENCHMARK

Because of our limited computation budget, we cannot afford to make systematic quantitative comparisons between the big learning and existing methods on pretraining a foundation model with large-scale data. Alternatively, we empirically reveal that the big learning is a superior fine-tuning strategy than the naive one.

Specifically, we initialize with the pretrained `xlnet-base-cased` model from the Hugging Face transformers library [51] and then test fine-tuning it on downstream RTE/MRPC/SST-2 tasks (from the GLUE Benchmark [48]) with ($i$) the naive fine-tuning strategy (termed FT) and ($ii$) the big learning (termed big-learn), respectively. Table 2 summarizes the quantitative evaluation results, where it's clear that big-learn consistently outperforms FT, even without careful tuning. See Appendix F for details.

## 5 CONCLUSIONS

We propose the big learning that exhaustively exploits the available data information and potentially delivers all joint, conditional, and marginal sampling data capabilities. We reveal that the big learning ($i$) comes with extraordinary training flexibilities for complete/incomplete data and for customizing training tasks, ($ii$) contains most objectives of foundation models as special cases, and ($iii$) is a new dimension for upgrading conventional machine learning paradigms; we present the upgraded BigLearn-GAN as a demonstration example. Diverse experiments are conducted to justify the effectiveness of the presented big learning.

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

# Appendix of Big Learning

**Anonymous Authors**

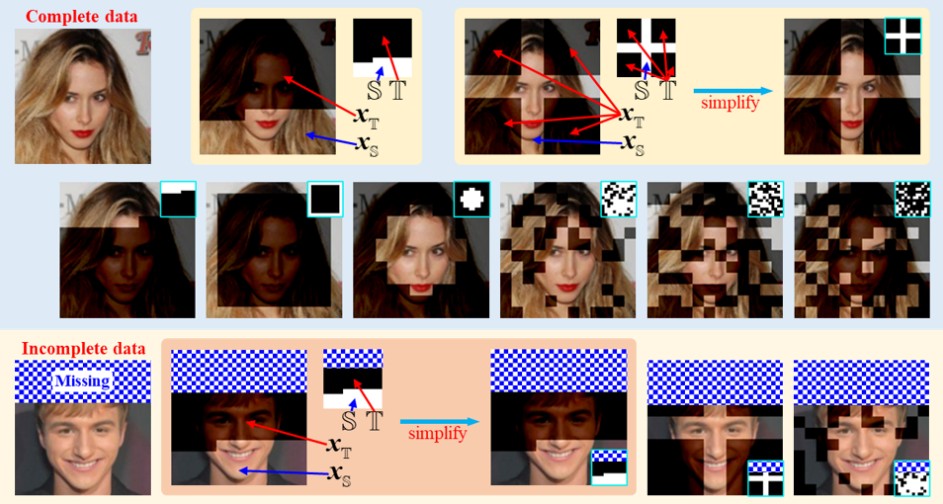

**(a)** A data can be exploited from many perspectives. When given a complete/incomplete data sample $\boldsymbol{x} \sim q(\boldsymbol{x})$, one simultaneously receives multiple joint, conditional, and marginal samples from $q(\boldsymbol{x}_\mathbb{T}|\boldsymbol{x}_\mathbb{S}), \forall (\mathbb{S}, \mathbb{T})$. These samples contain valuable data information associated with *e.g.,* data manifold and correlation among data patches (or words in text applications). Since they all demonstrate the *unique* underlying data distribution $q(\boldsymbol{x})$ (despite from diverse different perspectives), there is room with potential for introducing implicit regularizations among them via consistent multi-task training, *i.e.,* the big learning.

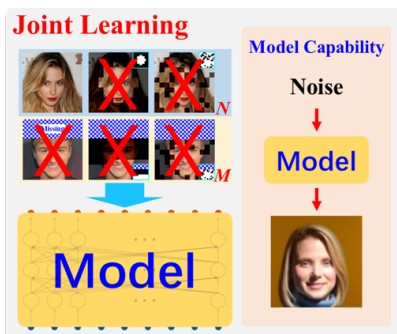

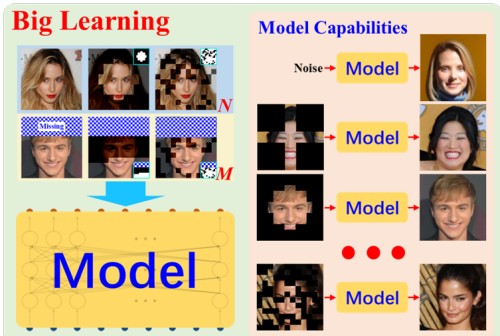

**(b)** The conventional machine learning, *i.e.,* joint learning with only the complete data, cannot fully exploit the data information, *e.g.,* the diverse correlations among data patches within conditional data and those within the incomplete data samples. Accordingly, only single joint data capability can be learned by the model.

**(c)** The (uni-modal) big learning flexibly and comprehensively exploits the diverse joint, conditional, and marginal samples inherent in complete and incomplete training data, leading to a consistent, unified, and principled learning framework underlying most foundation models. Besides, the big learning naturally delivers many/all joint, conditional, and marginal data capabilities across potentially diverse domains without computational overhead.

Figure 6: Big picture of the big learning, exampled by its uni-modal case.

## A  ON NAIVE MODELING OF ALL JOINT, CONDITIONAL, AND MARGINAL DATA DISTRIBUTIONS

We present with the unsupervised settings, where $\boldsymbol{x} \in \mathbb{R}^{L \times D}$ with length $L$ and dimension $D$ (like $L$ flattened patches of an image or $L$ words with $D = 1$). It's straightforward to generalize the following analyses to the general settings with a data sample $\boldsymbol{X} = (\boldsymbol{y}, \boldsymbol{x})$ contains an additional supervision $\boldsymbol{y} \in \mathbb{R}^{L^{\boldsymbol{y}} \times D^{\boldsymbol{y}}}$. Considering $D > 1$ and $D = 1$ for image patches and text words, respectively, we concentrate on analyzing the modeling of all joint, conditional, and marginal data distributions *w.r.t.* the length $L$ below.

As mentioned in the main manuscript, one need construct $N_{\text{all}} = \sum_{i=0}^{L-1} C_L^i (\sum_{k=1}^{L-i} C_{L-i}^k)$ models to naively model all joint, conditional, and marginal data distributions, to collect all joint, conditional, and marginal data capabilities. $C_L^i$ denotes the number of $i$-combinations from a set with $L$ elements.

To elaborate on that, consider a simple 3-length 1-dimensional problem with $\boldsymbol{x} = [x_1, x_2, x_3]^T$, where $L = 3$, $D = 1$, $x_i \in \mathbb{R}$, and the length index set $\mathbb{L} = \{1, 2, 3\}$.

- The goal of the joint matching is to deliver $p_{\boldsymbol{\theta}}(\boldsymbol{x}) \longrightarrow q(\boldsymbol{x})$ with one model $p_{\boldsymbol{\theta}}(\boldsymbol{x})$.
- By contrast, to naively model all joint, conditional, and marginal data distributions, one need construct 19 models for such a simple 3-length problem, *i.e.,*

$$p_{\boldsymbol{\theta}^1}(x_1),\ p_{\boldsymbol{\theta}^2}(x_2),\ p_{\boldsymbol{\theta}^3}(x_3),\ p_{\boldsymbol{\theta}^4}(x_1, x_2),\ p_{\boldsymbol{\theta}^5}(x_2, x_3),\ p_{\boldsymbol{\theta}^6}(x_1, x_3),\ p_{\boldsymbol{\theta}^7}(x_1, x_2, x_3),$$
$$p_{\boldsymbol{\theta}^8}(x_2|x_1),\ p_{\boldsymbol{\theta}^9}(x_3|x_1),\ p_{\boldsymbol{\theta}^{10}}(x_2, x_3|x_1),$$
$$p_{\boldsymbol{\theta}^{11}}(x_1|x_2),\ p_{\boldsymbol{\theta}^{12}}(x_3|x_2),\ p_{\boldsymbol{\theta}^{13}}(x_1, x_3|x_2), \tag{9}$$
$$p_{\boldsymbol{\theta}^{14}}(x_1|x_3),\ p_{\boldsymbol{\theta}^{15}}(x_2|x_3),\ p_{\boldsymbol{\theta}^{16}}(x_1, x_2|x_3),$$
$$p_{\boldsymbol{\theta}^{17}}(x_1|x_2, x_3),\ p_{\boldsymbol{\theta}^{18}}(x_2|x_1, x_3),\ p_{\boldsymbol{\theta}^{19}}(x_3|x_1, x_2).$$

Based on the above 3-length problem, one can readily summarize the following two steps in calculating the number of models in naively modeling all joint, conditional, and marginal data distributions, *i.e.,* $q(\boldsymbol{x}_{\mathbb{T}}|\boldsymbol{x}_{\mathbb{S}}), \forall \mathbb{S} \subset \mathbb{L}, \mathbb{T} \subseteq \mathbb{L}, \mathbb{T} \neq \emptyset$.

1. **Sample $\mathbb{S}$.** The source index set $\mathbb{S}$ may contain $\{0, \cdots, L-1\}$ indexes/locations, where $\mathbb{S}$ containing 0 index corresponds to joint/marginal generations and $\mathbb{S}$ containing $\geq 1$ indexes corresponds to conditional generations/completions. For a special case with $i$ indexes in $\mathbb{S}$ with $i \in [0, L-1]$, one has $C_L^i$ ways to specify that source index set $\mathbb{S}$.
2. **Sample $\mathbb{T}$ conditioned on $\mathbb{S}$.** Given a $\mathbb{S}$ consisting of $i$ indexes, the target index set $\mathbb{T}$ could contain $\{1, \cdots, L-i\}$ indexes/locations outside $\mathbb{S}$. For a special case of $\mathbb{T}$ containing $k$ indexes where $k \in [1, L-i]$, one has $C_{L-i}^k$ ways to specify the target $\mathbb{T}$.

Therefore, to naively model all joint, conditional, and marginal data distributions, one need construct $N_{\text{all}} = \sum_{i=0}^{L-1} C_L^i (\sum_{k=1}^{L-i} C_{L-i}^k)$ models, which, however, is prohibitive in practice.

Note with ideal modeling of $q(\boldsymbol{x}_{\mathbb{T}}|\boldsymbol{x}_{\mathbb{S}})$, the orders in $\mathbb{S}/\mathbb{T}$ should not matter. However, that may not hold true considering practical constraints, *e.g.,* where existing joint matching techniques fail to model the multi-mode characteristics of $\boldsymbol{x}_{\mathbb{T}}$. Besides, in the NLP application of language modeling, one may be interested in versatile (conditional) generation ordering (as defined in $\mathbb{T}$), mimicking the permutation language modeling [52]. In that case, to naively modeling all joint, conditional, and marginal data distributions, one need construct $N'_{\text{all}} = \sum_{i=0}^{L-1} C_L^i (\sum_{k=1}^{L-i} A_{L-i}^k)$ models to take into consideration the order of $\mathbb{T}$, where the order of $\mathbb{S}$ is ignored and $A_{L-i}^k$ denotes the number of the ordered arrangements of $k$ elements from a set with $L - 1$ elements. Similarly, one need construct $N''_{\text{all}} = \sum_{i=0}^{L-1} A_L^i (\sum_{k=1}^{L-i} A_{L-i}^k)$ models to model the orders in both $\mathbb{S}$ and $\mathbb{T}$.

## B  DERIVATIONS OF THE GAN EXAMPLE ASSOCIATED WITH EQS. (4) AND (5)

Here we present the detailed derivations/proofs for the GAN example associated with Eqs. (4) and (5) of the main manuscript. For better understanding, we begin with a simplified case where $\mathbb{T} = \mathbb{L}\backslash\mathbb{S}$, followed by generalizing the results to the general situations with $\mathbb{T} \subseteq \mathbb{L}\backslash\mathbb{S}$.

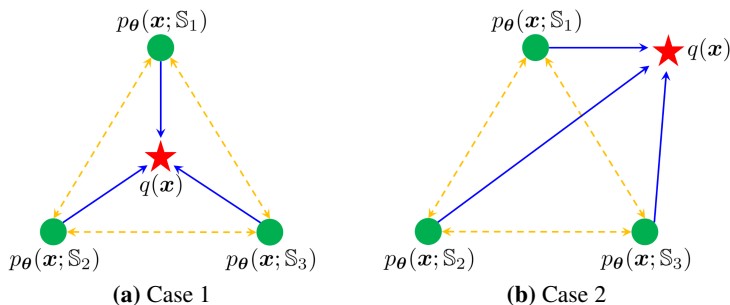

**Figure 7:** Demonstration of unsupervised big learning based on GANs.

## B.1 $\mathbb{T} = \mathbb{L}\backslash\mathbb{S}$

To leverage the GAN training framework [15], one needs the sampling capabilities from the distributions of interest. With $\mathbb{T} = \mathbb{L}\backslash\mathbb{S}$, here we are interested in the joint distributions with accessible sampling capabilities, including

$$q(\boldsymbol{x})$$
$$p_{\boldsymbol{\theta}}(\boldsymbol{x};\mathbb{S}) = p_{\boldsymbol{\theta}}(\boldsymbol{x}_{\mathbb{L}\backslash\mathbb{S}}|\boldsymbol{x}_{\mathbb{S}})q(\boldsymbol{x}_{\mathbb{S}}) \quad \forall\mathbb{S}. \tag{10}$$

Note one can of course exploit the flexibility of big learning to define other joint distributions with sampling capabilities, such as an recursively defined distribution

$$p_{\boldsymbol{\theta}}(\boldsymbol{x};\mathbb{S}^1,\mathbb{S}^2) = p_{\boldsymbol{\theta}}(\boldsymbol{x}_{\mathbb{L}\backslash\mathbb{S}^2}|\boldsymbol{x}_{\mathbb{S}^2})p_{\boldsymbol{\theta}}(\boldsymbol{x}_{\mathbb{S}^2}), \tag{11}$$

where $p_{\boldsymbol{\theta}}(\boldsymbol{x}_{\mathbb{S}^2}) = \int p_{\boldsymbol{\theta}}(\boldsymbol{x}_{\mathbb{L}\backslash\mathbb{S}^1}|\boldsymbol{x}_{\mathbb{S}^1})q(\boldsymbol{x}_{\mathbb{S}^1})d\boldsymbol{x}_{\mathbb{L}\backslash\mathbb{S}^2}$. For simplicity, we focus on the simplified settings in (10) and leave the interesting but complicated recursive case for future research.

Given the underlying data distribution $q(\boldsymbol{x})$ and "model" distributions $p_{\boldsymbol{\theta}}(\boldsymbol{x};\mathbb{S})$ in (10),

1. one can match any $p_{\boldsymbol{\theta}}(\boldsymbol{x};\mathbb{S})$ to $q(\boldsymbol{x})$ adversarially with a GAN. Take the standard GAN [15] for an example, the objective is

$$\min_{\boldsymbol{\theta}}\max_{\boldsymbol{\phi}} \mathbb{E}_{q(\boldsymbol{x})}\log\sigma(f_{\boldsymbol{\phi}}(\boldsymbol{x};\mathbb{S})) + \mathbb{E}_{p_{\boldsymbol{\theta}}(\boldsymbol{x}_{\mathbb{L}\backslash\mathbb{S}}|\boldsymbol{x}_{\mathbb{S}})q(\boldsymbol{x}_{\mathbb{S}})}\log(1-\sigma(f_{\boldsymbol{\phi}}(\boldsymbol{x};\mathbb{S}))), \tag{12}$$

where the optimal $f_{\boldsymbol{\phi}^*}(\boldsymbol{x};\mathbb{S}) = \log\frac{q(\boldsymbol{x})}{p_{\boldsymbol{\theta}}(\boldsymbol{x}_{\mathbb{L}\backslash\mathbb{S}}|\boldsymbol{x}_{\mathbb{S}})q(\boldsymbol{x}_{\mathbb{S}})} = \log\frac{q(\boldsymbol{x}_{\mathbb{L}\backslash\mathbb{S}}|\boldsymbol{x}_{\mathbb{S}})}{p_{\boldsymbol{\theta}}(\boldsymbol{x}_{\mathbb{L}\backslash\mathbb{S}}|\boldsymbol{x}_{\mathbb{S}})}$. Ideally, optimizing the above objective is identical to minimizing the Jensen-Shannon divergence $\mathrm{JS}[q(\boldsymbol{x})\|p_{\boldsymbol{\theta}}(\boldsymbol{x};\mathbb{S})]$, as illustrated with the blue solid arrows in Fig. 7.

2. one can also conduct matching among any two model distributions (*e.g.,* $p_{\boldsymbol{\theta}}(\boldsymbol{x};\mathbb{S}^1) = p_{\boldsymbol{\theta}}(\boldsymbol{x}_{\mathbb{L}\backslash\mathbb{S}^1}|\boldsymbol{x}_{\mathbb{S}^1})q(\boldsymbol{x}_{\mathbb{S}^1})$ and $p_{\boldsymbol{\theta}}(\boldsymbol{x};\mathbb{S}^2) = p_{\boldsymbol{\theta}}(\boldsymbol{x}_{\mathbb{L}\backslash\mathbb{S}^2}|\boldsymbol{x}_{\mathbb{S}^2})q(\boldsymbol{x}_{\mathbb{S}^2})$) to enable communications/cooperations among them, via optimizing

$$\min_{\boldsymbol{\theta}}\max_{\boldsymbol{\phi}} \begin{cases} \mathbb{E}_{p_{\boldsymbol{\theta}}(\boldsymbol{x}_{\mathbb{L}\backslash\mathbb{S}^1}|\boldsymbol{x}_{\mathbb{S}^1})q(\boldsymbol{x}_{\mathbb{S}^1})}\log\sigma(f'_{\boldsymbol{\phi}}(\boldsymbol{x};\mathbb{S}^1,\mathbb{S}^2)) \\ + \mathbb{E}_{p_{\boldsymbol{\theta}}(\boldsymbol{x}_{\mathbb{L}\backslash\mathbb{S}^2}|\boldsymbol{x}_{\mathbb{S}^2})q(\boldsymbol{x}_{\mathbb{S}^2})}\log(1-\sigma(f'_{\boldsymbol{\phi}}(\boldsymbol{x};\mathbb{S}^1,\mathbb{S}^2))) \end{cases} \tag{13}$$

where the optimal $f'_{\boldsymbol{\phi}^*}(\boldsymbol{x};\mathbb{S}^1,\mathbb{S}^2) = \log\frac{p_{\boldsymbol{\theta}}(\boldsymbol{x}_{\mathbb{L}\backslash\mathbb{S}^1}|\boldsymbol{x}_{\mathbb{S}^1})q(\boldsymbol{x}_{\mathbb{S}^1})}{p_{\boldsymbol{\theta}}(\boldsymbol{x}_{\mathbb{L}\backslash\mathbb{S}^2}|\boldsymbol{x}_{\mathbb{S}^2})q(\boldsymbol{x}_{\mathbb{S}^2})}$. The orange dotted arrows in Fig. 7 demonstrate such idea.

At first sight of Eqs. (12) and (13), it seems one should at least construct two discriminators, with $f_{\boldsymbol{\phi}}(\boldsymbol{x};\mathbb{S})$ and $f'_{\boldsymbol{\phi}}(\boldsymbol{x};\mathbb{S}^1,\mathbb{S}^2)$ respectively. However, we notice that

$$f'_{\boldsymbol{\phi}^*}(\boldsymbol{x};\mathbb{S}^1,\mathbb{S}^2) = \log\frac{q(\boldsymbol{x})}{p_{\boldsymbol{\theta}}(\boldsymbol{x}_{\mathbb{L}\backslash\mathbb{S}^2}|\boldsymbol{x}_{\mathbb{S}^2})q(\boldsymbol{x}_{\mathbb{S}^2})} - \log\frac{q(\boldsymbol{x})}{p_{\boldsymbol{\theta}}(\boldsymbol{x}_{\mathbb{L}\backslash\mathbb{S}^1}|\boldsymbol{x}_{\mathbb{S}^1})q(\boldsymbol{x}_{\mathbb{S}^1})}$$
$$= f_{\boldsymbol{\phi}^*}(\boldsymbol{x};\mathbb{S}^2) - f_{\boldsymbol{\phi}^*}(\boldsymbol{x};\mathbb{S}^1).$$

Accordingly, we propose to employ further simplification that builds $f'_\phi(\boldsymbol{x}; \mathbb{S}^1, \mathbb{S}^2)$ on top of $f_\phi(\boldsymbol{x}; \mathbb{S})$, *i.e.,* we reformulate (13) as

$$\min_{\boldsymbol{\theta}} \max_{\phi} \begin{cases} \mathbb{E}_{p_{\boldsymbol{\theta}}(\boldsymbol{x}_{\mathbb{L}\setminus\mathbb{S}^1}|\boldsymbol{x}_{\mathbb{S}^1})q(\boldsymbol{x}_{\mathbb{S}^1})}\log\sigma[f_\phi(\boldsymbol{x}; \mathbb{S}^2) - f_\phi(\boldsymbol{x}; \mathbb{S}^1)] \\ + \mathbb{E}_{p_{\boldsymbol{\theta}}(\boldsymbol{x}_{\mathbb{L}\setminus\mathbb{S}^2}|\boldsymbol{x}_{\mathbb{S}^2})q(\boldsymbol{x}_{\mathbb{S}^2})}\log\sigma[f_\phi(\boldsymbol{x}; \mathbb{S}^1) - f_\phi(\boldsymbol{x}; \mathbb{S}^2)]. \end{cases} \tag{14}$$

Till now, we present the derivations associated with $\mathbb{T} = \mathbb{L}\setminus\mathbb{S}$, *i.e.,* matching in the joint space. In what follows, we generalize to the settings with $\mathbb{T} \subseteq \mathbb{L}\setminus\mathbb{S}$, to deliver (unsupervised) big learning in all joint, conditional, and marginal spaces.

## B.2 $\mathbb{T} \subseteq \mathbb{L}\setminus\mathbb{S}$

Similar to the previous section, we also consider simplified situations with no recursiveness, that is, we do not consider a model distribution $p_{\boldsymbol{\theta}}(\boldsymbol{x}_{\mathbb{T}}|\boldsymbol{x}_{\mathbb{S}})p_{\boldsymbol{\theta}}(\boldsymbol{x}_{\mathbb{S}})$, even though such recursive flexibility of big learning is quite interesting. We leave that as future research.

Accordingly, the considered joint, conditional, and marginal distributions with sampling capabilities are

$$\begin{aligned} &q(\boldsymbol{x}_{\mathbb{S}\cup\mathbb{T}}) \\ &p_{\boldsymbol{\theta}}(\boldsymbol{x}_{\mathbb{S}\cup\mathbb{T}}) = p_{\boldsymbol{\theta}}(\boldsymbol{x}_{\mathbb{T}}|\boldsymbol{x}_{\mathbb{S}})q(\boldsymbol{x}_{\mathbb{S}}) \qquad \forall \mathbb{S}, \mathbb{T} \end{aligned} \tag{15}$$

where $\mathbb{S} \cup \mathbb{T}$ need not be $\mathbb{L}$. Note $\mathbb{S} \cup \mathbb{T} \subset \mathbb{L}$ means the corresponding $q(\boldsymbol{x}_{\mathbb{S}\cup\mathbb{T}})$ is a *marginal* data distribution, whose data samples are readily accessible from those of $q(\boldsymbol{x})$.

Similar to the previous section,

- one can match any model distribution $p_{\boldsymbol{\theta}}(\boldsymbol{x}_{\mathbb{S}\cup\mathbb{T}})$ to the underlying joint/marginal data distribution $q(\boldsymbol{x}_{\mathbb{S}\cup\mathbb{T}})$, via the standard GAN objective

$$\min_{\boldsymbol{\theta}} \max_{\phi} \mathbb{E}_{q(\boldsymbol{x}_{\mathbb{S}\cup\mathbb{T}})} \log \sigma(f_\phi(\boldsymbol{x}; \mathbb{S}, \mathbb{T})) + \mathbb{E}_{p_{\boldsymbol{\theta}}(\boldsymbol{x}_{\mathbb{T}}|\boldsymbol{x}_{\mathbb{S}})q(\boldsymbol{x}_{\mathbb{S}})} \log(1 - \sigma(f_\phi(\boldsymbol{x}; \mathbb{S}, \mathbb{T}))), \tag{16}$$

  where $f_{\phi^*}(\boldsymbol{x}; \mathbb{S}, \mathbb{T}) = \log \frac{q(\boldsymbol{x}_{\mathbb{S}\cup\mathbb{T}})}{p_{\boldsymbol{\theta}}(\boldsymbol{x}_{\mathbb{T}}|\boldsymbol{x}_{\mathbb{S}})q(\boldsymbol{x}_{\mathbb{S}})} = \log \frac{q(\boldsymbol{x}_{\mathbb{T}}|\boldsymbol{x}_{\mathbb{S}})}{p_{\boldsymbol{\theta}}(\boldsymbol{x}_{\mathbb{T}}|\boldsymbol{x}_{\mathbb{S}})}$.

- one can also conduct matching among any two model distributions, *e.g.,* $p_{\boldsymbol{\theta}}(\boldsymbol{x}_{\mathbb{T}^1}|\boldsymbol{x}_{\mathbb{S}^1})q(\boldsymbol{x}_{\mathbb{S}^1})$ and $p_{\boldsymbol{\theta}}(\boldsymbol{x}_{\mathbb{T}^2}|\boldsymbol{x}_{\mathbb{S}^2})q(\boldsymbol{x}_{\mathbb{S}^2})$, as long as $\mathbb{S}^1 \cup \mathbb{T}^1 = \mathbb{S}^2 \cup \mathbb{T}^2$, with the corresponding objective

$$\min_{\boldsymbol{\theta}} \max_{\phi} \begin{cases} \mathbb{E}_{p_{\boldsymbol{\theta}}(\boldsymbol{x}_{\mathbb{T}^1}|\boldsymbol{x}_{\mathbb{S}^1})q(\boldsymbol{x}_{\mathbb{S}^1})}\log\sigma(f_\phi(\boldsymbol{x}; \mathbb{S}^1, \mathbb{T}^1, \mathbb{S}^2, \mathbb{T}^2)) \\ + \mathbb{E}_{p_{\boldsymbol{\theta}}(\boldsymbol{x}_{\mathbb{T}^2}|\boldsymbol{x}_{\mathbb{S}^2})q(\boldsymbol{x}_{\mathbb{S}^2})}\log(1 - \sigma(f_\phi(\boldsymbol{x}; \mathbb{S}^1, \mathbb{T}^1, \mathbb{S}^2, \mathbb{T}^2))), \end{cases} \tag{17}$$

  where $f'_{\phi^*}(\boldsymbol{x}; \mathbb{S}^1, \mathbb{T}^1, \mathbb{S}^2, \mathbb{T}^2) = \log \frac{p_{\boldsymbol{\theta}}(\boldsymbol{x}_{\mathbb{T}^1}|\boldsymbol{x}_{\mathbb{S}^1})q(\boldsymbol{x}_{\mathbb{S}^1})}{p_{\boldsymbol{\theta}}(\boldsymbol{x}_{\mathbb{T}^2}|\boldsymbol{x}_{\mathbb{S}^2})q(\boldsymbol{x}_{\mathbb{S}^2})}$.

  For further simplifications, we again resort to

$$\begin{aligned} f'_{\phi^*}(\boldsymbol{x}; \mathbb{S}^1, \mathbb{T}^1, \mathbb{S}^2, \mathbb{T}^2) &= \log \frac{q(\boldsymbol{x}_{\mathbb{S}^2\cup\mathbb{T}^2})}{p_{\boldsymbol{\theta}}(\boldsymbol{x}_{\mathbb{T}^2}|\boldsymbol{x}_{\mathbb{S}^2})q(\boldsymbol{x}_{\mathbb{S}^2})} - \log \frac{q(\boldsymbol{x}_{\mathbb{S}^1\cup\mathbb{T}^1})}{p_{\boldsymbol{\theta}}(\boldsymbol{x}_{\mathbb{T}^1}|\boldsymbol{x}_{\mathbb{S}^1})q(\boldsymbol{x}_{\mathbb{S}^1})} \\ &= f_{\phi^*}(\boldsymbol{x}; \mathbb{S}^2, \mathbb{T}^2) - f_{\phi^*}(\boldsymbol{x}; \mathbb{S}^1, \mathbb{T}^1) \end{aligned}$$

  and build $f'_\phi(\boldsymbol{x}; \mathbb{S}^1, \mathbb{T}^1, \mathbb{S}^2, \mathbb{T}^2)$ on top of $f_\phi(\boldsymbol{x}; \mathbb{S}, \mathbb{T})$.

  Accordingly, Eq. (17) is reformulated as

$$\min_{\boldsymbol{\theta}} \max_{\phi} \begin{cases} \mathbb{E}_{p_{\boldsymbol{\theta}}(\boldsymbol{x}_{\mathbb{T}^1}|\boldsymbol{x}_{\mathbb{S}^1})q(\boldsymbol{x}_{\mathbb{S}^1})}\log\sigma[f_\phi(\boldsymbol{x}; \mathbb{S}^2, \mathbb{T}^2) - f_\phi(\boldsymbol{x}; \mathbb{S}^1, \mathbb{T}^1)] \\ + \mathbb{E}_{p_{\boldsymbol{\theta}}(\boldsymbol{x}_{\mathbb{T}^2}|\boldsymbol{x}_{\mathbb{S}^2})q(\boldsymbol{x}_{\mathbb{S}^2})}\log\sigma[f_\phi(\boldsymbol{x}; \mathbb{S}^1, \mathbb{T}^1) - f_\phi(\boldsymbol{x}; \mathbb{S}^2, \mathbb{T}^2)]. \end{cases} \tag{18}$$

Accordingly, we conclude the proofs for the GAN example of the main manuscript.

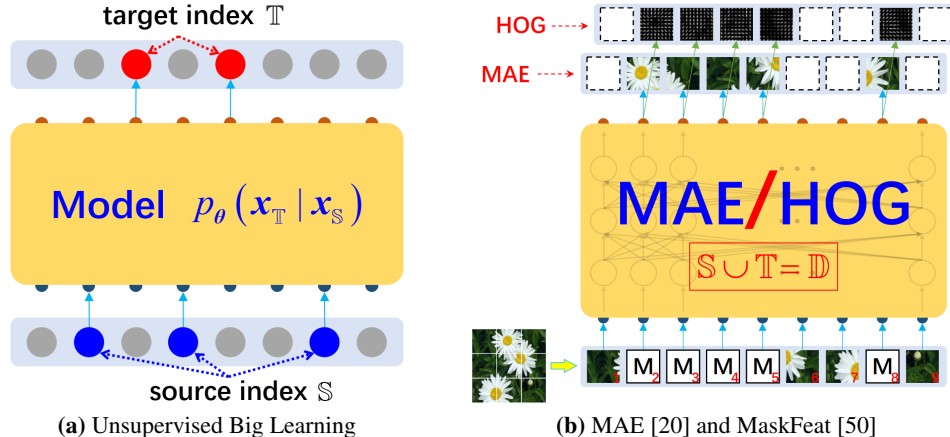

**(a)** Unsupervised Big Learning  **(b)** MAE [20] and MaskFeat [50]

Figure 8: Unsupervised big learning (a) and its special cases (b). Often a mask token [M] is inserted to the input locations outside $\mathbb{S}$ for forward propagation, while no loss is back-propagated to the output locations outside $\mathbb{T}$. Note inserting the [M] tokens later in a middle layer (but at the same location) often lightens the computation and memory burdens but improves the performance [20].

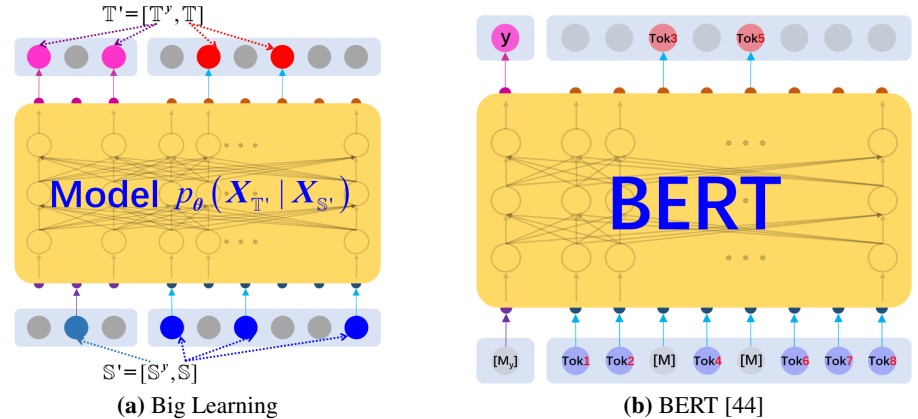

**(a)** Big Learning  **(b)** BERT [44]

Figure 9: Big learning (a) and its special case of BERT (b). Similar to the mask token [M] for $x$ (see Fig. 8b), we employ another mask token [$M_y$] for $y$, which works identically to the classification token [CLS] in BERT settings [44] and the start-of-sentence token in GPT settings [6]. Often inserting [M]/[$M_y$] tokens later in a middle layer improves performance [20; 46].

## C  ON MODEL ARCHITECTURES OF THE GAN EXAMPLE IN EQS. (4) AND (5)

We next focus on discussing the model architectures of the GAN generator and discriminator employed in Eqs. (16) and (18) (*i.e.,* Eqs. (4) and (5) of the main manuscript).

Recently, the community begins to exploit integrating ViTs into GANs [22; 31; 55; 54]. For example, the ViTGAN [31], delivering SOTA generative performance, employs simple modifications to the ViT architecture to construct the generator and the discriminator, but adopts *many* techniques to regularize the ViT-based discriminator for stable training. Motivated by the modeling flexibility of ViTs, we also employ ViT-based GAN generator and discriminator in the experiments, but similarly, find it challenging to stabilize GAN training with a ViT-based discriminator. It's worth highlighting that it's possible to design other alternative model architectures for the big learning; we employ what's presented below for a demonstration.

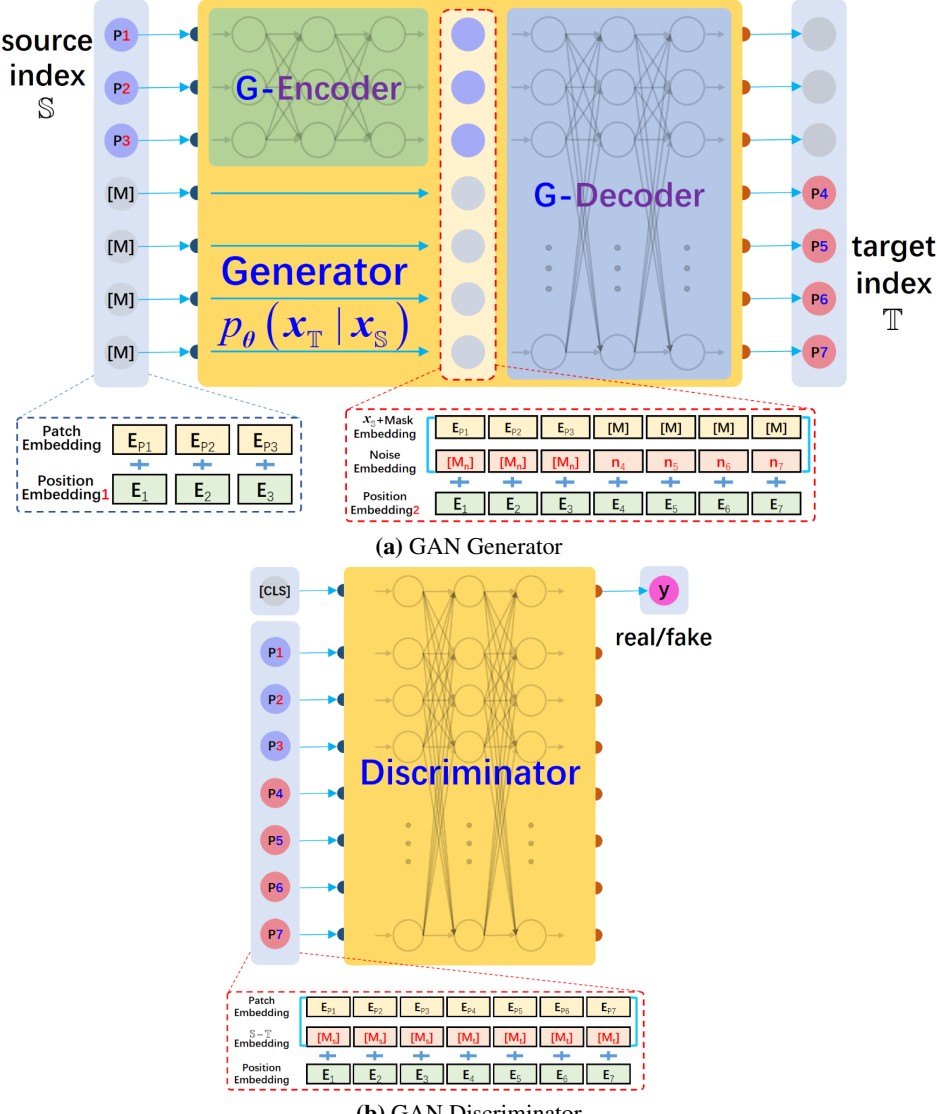

**(a)** GAN Generator

**(b)** GAN Discriminator

Figure 10: Example implementations of the GAN generator and discriminator employed in Eqs. (16) and (18) (*i.e.,* Eqs. (4) and (5) of the main manuscript).

Fig. 10 demonstrates the employed GAN generator and discriminator, both of which are constructed with Transformers/ViTs to exploit their modeling capabilities and flexibilities.

- **GAN Generator.** Following the MAE [20], we design the GAN generator $p_{\boldsymbol{\theta}}(\boldsymbol{x}_{\mathbb{T}}|\boldsymbol{x}_{\mathbb{S}})$ with an autoencoder-like architecture, which employs an encoding G-Encoder and a decoding G-Decoder, as shown in Fig. 10a. The G-Encoder encodes the source patches $\boldsymbol{x}_{\mathbb{S}}$ (if any) to their latent codes; then, these codes are combined with the mask tokens [M], patch-wise noise embeddings, and new positional encodings to serve as the input of the G-Decoder; finally, the G-Decoder transforms its input to generate the target patches $\boldsymbol{x}_{\mathbb{T}}$.

  [M] tokens are inserted later in a middle layer, because doing this often improves performance and lowers the computational burden [46; 20]. A noise $\boldsymbol{z}$ is mapped with an 8-layer MLP to produce the patch-wise noise embeddings $\{\boldsymbol{n}_1, \cdots, \boldsymbol{n}_L\}$. Note we also introduce another toke [$M_n$] to indicate no noise embeddings are necessary at the corresponding source locations in $\mathbb{S}$.

---

**Algorithm 1** Big Learning (exampled by the uni-modal big learning in Eq. 3)

---

**Input:** Training data, maximum number $N$ of iterations, $q(\mathbb{S}, \mathbb{T})$ that defines the sampling of $(\mathbb{S}, \mathbb{T}) \in \mathbf{\Omega}$, and an application-dependent loss function $L(\boldsymbol{\theta}) = \mathcal{D}[p_{\boldsymbol{\theta}}(\boldsymbol{x}_{\mathbb{T}}|\boldsymbol{x}_{\mathbb{S}})||q(\boldsymbol{x}_{\mathbb{T}}|\boldsymbol{x}_{\mathbb{S}})]$
**Output:** A consistent local optimum $\boldsymbol{\theta}^*$
 1: Randomly initialize $\boldsymbol{\theta}$
 2: **while** iter $\leq N$ **do**
 3:     Sample a $(\mathbb{S}, \mathbb{T})$ pair from $q(\mathbb{S}, \mathbb{T})$
 4:     Calculate the loss $L(\boldsymbol{\theta})$ that encourages the matching $p_{\boldsymbol{\theta}}(\boldsymbol{x}_{\mathbb{T}}|\boldsymbol{x}_{\mathbb{S}}) \longrightarrow q(\boldsymbol{x}_{\mathbb{T}}|\boldsymbol{x}_{\mathbb{S}})$
 5:     Update $\boldsymbol{\theta} \leftarrow \boldsymbol{\theta} - \nabla_{\boldsymbol{\theta}} L(\boldsymbol{\theta})$
 6: **end while**

---

**Algorithm 2** Big Learning Generative Adversarial Nets (BigLearn-GAN)

---

**Input:** Training data, maximum number $N$ of iterations.
**Output:** Consistent local optima, the generator $\boldsymbol{\theta}^*$ and discriminator $\boldsymbol{\phi}^*$.
 1: Randomly initialize $\boldsymbol{\phi}$ and $\boldsymbol{\theta}$
 2: **while** iter $\leq N$ **do**
 3:     Sample a $(\mathbb{S}^1, \mathbb{T}^1)$ pair from $q(\mathbb{S}, \mathbb{T})$
 4:     Sample $\mathbb{S}^2$ from $\mathbb{S}^1 \cup \mathbb{T}^1$ and then set $\mathbb{T}^2 = \mathbb{S}^1 \cup \mathbb{T}^1 - \mathbb{S}^2$
 5:     # Update the discriminator parameters $\boldsymbol{\phi}$
 6:         # Calculate model-to-data losses based on Eq. 4
 7:             $(i)$ Calculate the $1^{st}$ discriminator loss $J_1(\boldsymbol{\phi})$ based on $(\mathbb{S}^1, \mathbb{T}^1)$
 8:             $(ii)$ Calculate the $2^{ed}$ discriminator loss $J_2(\boldsymbol{\phi})$ based on $(\mathbb{S}^2, \mathbb{T}^2)$
 9:         # Calculate the model-to-model communication loss based on Eq. 5
10:             $(iii)$ Calculate the $3^{rd}$ discriminator loss $J_3(\boldsymbol{\phi})$ based on both $(\mathbb{S}^1, \mathbb{T}^1)$ and $(\mathbb{S}^2, \mathbb{T}^2)$
11:         Update $\boldsymbol{\phi} \leftarrow \boldsymbol{\phi} - \nabla_{\boldsymbol{\phi}}[J_1(\boldsymbol{\phi}) + J_2(\boldsymbol{\phi}) + J_3(\boldsymbol{\phi})]$
12:                     ▷ `often regularized by the gradient penalty [34]`
13:     # Update the generator parameters $\boldsymbol{\theta}$
14:         # Calculate model-to-data losses based on Eq. 4
15:             $(i)$ Calculate the $1^{st}$ generator loss $L_1(\boldsymbol{\theta})$ based on $(\mathbb{S}^1, \mathbb{T}^1)$
16:             $(ii)$ Calculate the $2^{ed}$ generator loss $L_2(\boldsymbol{\theta})$ based on $(\mathbb{S}^2, \mathbb{T}^2)$
17:         # Calculate the model-to-model communication loss based on Eq. 5
18:             $(iii)$ Calculate the $3^{rd}$ generator loss $L_3(\boldsymbol{\theta})$ based on both $(\mathbb{S}^1, \mathbb{T}^1)$ and $(\mathbb{S}^2, \mathbb{T}^2)$
19:         Update $\boldsymbol{\theta} \leftarrow \boldsymbol{\theta} - \nabla_{\boldsymbol{\theta}}[L_1(\boldsymbol{\theta}) + L_2(\boldsymbol{\theta}) + L_3(\boldsymbol{\theta})]$
20: **end while**

---

- **GAN Discriminator.** As shown in Fig. 10b, we also modify the Transformer/ViT architecture to construct the universal GAN discriminator $\sigma(f_{\boldsymbol{\phi}}(\boldsymbol{x}; \mathbb{S}, \mathbb{T}))$ that applies to all $(\mathbb{S}, \mathbb{T})$ cases. We employ an additional `CLS` token mimicking the BERT, whose output indicates whether the input patches are realistic or not (more specifically, whether they form a "real" data from $q(\boldsymbol{x}_{\mathbb{S} \cup \mathbb{T}})$ or a fake one from $p_{\boldsymbol{\theta}}(\boldsymbol{x}_{\mathbb{T}}|\boldsymbol{x}_{\mathbb{S}})q(\boldsymbol{x}_{\mathbb{S}})$, by referring to (16)). The input of the discriminator consists of patch embeddings, positional embeddings, and two new special tokens (`[M`$_s$`]` and `[M`$_t$`]`) that indicate source or target patches mimicking the sentence tokens in the BERT.

## D   EXPERIMENTAL SETTINGS USED IN SECTIONS 4.1 AND 4.2 OF THE MAIN MANUSCRIPT

We employ the same model architectures in the previous Section C for the experiments on the MNIST and CelebA datasets, with the detailed hyperparameters summarized in Table 3. Despite the relatively small models used, we find that big learning is capable of delivering potentially all joint, conditional, and marginal data capabilities simultaneously. We adopt the AdamW optimizer [33] with $\beta = (0.1, 0.999)$ and constant learning rates for both the generator and the discriminator. Code will be released upon publication.

Overall, we find it's quite straightforward to implement the MNIST experiments with the standard implementations discussed in Sections B and C, without resorting to any "tricks" like warm-up or

Table 3: Hyperparameters used in the experiments.

| Dataset | MNIST | CelebA |
|---|---|---|
| Image size | 64 | 120 |
| Patch size | 8 | 10 |
| G-Encoder depth | 6 | 6 |
| G-Encoder #heads | 8 | 8 |
| G-Encoder dim | 256 | 256 |
| G-Decoder depth | 6 | 6 |
| G-Decoder #heads | 8 | 8 |
| G-Decoder dim | 512 | 512 |
| D depth | 6 | 6 |
| D #heads | 8 | 8 |
| D dim | 256 | 256 |
| GP [34] | real | real |
| $\lambda_{\text{GP}}$ | 10 | 10 |
| Learning rate | $10^{-4}$ | $10^{-4}$ |
| Batch size | 256 | 128 |
| Source ratio $\|\mathbb{S}^1\|/\|\mathbb{L}\|$ | Beta(0.5,3) | Beta(0.5,3) |
| Target ratio $\|\mathbb{T}^1\|/\|\mathbb{L}\backslash\mathbb{S}^1\|$ | Beta(3,0.5) | Beta(3,0.5) |
| Communication source ratio $\|\mathbb{S}^2\|/\|\mathbb{S}^1\cup\mathbb{T}^1\|$ | Beta(0.5,3) | Beta(0.5,3) |

gradient clipping. However, on the more complicated CelebA experiments, we find it's necessary to employ some, as detailed below.

- We employ warm-up in the first 10 epochs for both the GAN generator and discriminator; after that, we use the constant learning rate given in Table 3.

- We apply gradient clipping, with the max norm of 5, to both the generator and discriminator optimizers.

- Similar to Lee et al. [31], we also find it challenging to stabilize GAN training with a ViT-based discriminator. To deal with that, we additionally ($i$) overlap image patches [31] with *e.g.,* 2 pixels at the input of the discriminator (different from the non-overlapping image patches used in the vanilla ViT); and ($ii$) use a larger hyperparameter $\epsilon = 10^{-5}$ in the AdamW optimizer.

Other empirical experiences are listed below.

- We empirically find that the last normalization layers of both the GAN generator and discriminator have a significant influence on the learning stability and final performance. Specifically, replacing the last `LayerNorm` of the G-Decoder of the generator with a `LeakyReLU` leads to improved generative performance, whereas replacing the last `LayerNorm` of the discriminator with other normalization/activation layers results in training collapse.

- Employing an additional convolutional head (like a 3-layer CNN) to the output of the generator often leads to improved performance and training stability.

- Instead of only introducing noise embeddings at the first layer of the G-Decoder of the generator, as shown in Fig. 10a, we find it's beneficial to concatenate the same set of noise embeddings layer-wisely into the G-Decoder layers.

## E  BIG LEARNING UNIFIES CLASSIFICATION AND GENERATION

After following [3; 39] to vector-quantize an image into discrete tokens $\boldsymbol{x} \in \mathbb{Z}^{L\times 1}$, the observed random variable $\boldsymbol{X} = (y, \boldsymbol{x})$ with discrete label $y$ now has only one data type. Accordingly, one can readily generalize (6) of the uni-model unsupervised big learning to solve the problem.

Specifically, with a Transformer-based universal model $p_{\boldsymbol{\theta}}(\boldsymbol{X}_{\bar{\mathbb{T}}'}|\boldsymbol{X}_{\bar{\mathbb{S}}'})$ that models the generative process of a target token $\boldsymbol{X}_{\bar{\mathbb{T}}'}$ given source ones $\boldsymbol{X}_{\bar{\mathbb{S}}'}$ for *any* $(\bar{\mathbb{S}}', \bar{\mathbb{T}}')$ pair, the big learning yields

$$\max_{\boldsymbol{\theta}} \mathbb{E}_{q(\mathbb{S}',\mathbb{T}')} \sum_{(\bar{\mathbb{S}}',\bar{\mathbb{T}}') \in \Xi'_{\mathbb{S}',\mathbb{T}'}} \mathbb{E}_{q(\boldsymbol{X}_{\bar{\mathbb{T}}'}|\boldsymbol{X}_{\bar{\mathbb{S}}'})} \log p_{\boldsymbol{\theta}}(\boldsymbol{X}_{\bar{\mathbb{T}}'}|\boldsymbol{X}_{\bar{\mathbb{S}}'}), \tag{19}$$

where $q(\mathbb{S}', \mathbb{T}')$ denotes the sampling process of $(\mathbb{S}', \mathbb{T}')$ with random permutations, $\mathbb{T}' = \{t_1, t_2, \cdots\}$, $\Xi'_{\mathbb{S}',\mathbb{T}'} = \{(\mathbb{S}', t_1), (\{\mathbb{S}', t_1\}, t_2), (\{\mathbb{S}', t_1, t_2\}, t_3), \cdots\}$, often $p_{\boldsymbol{\theta}}(\boldsymbol{X}_{\bar{\mathbb{T}}'}|\boldsymbol{X}_{\bar{\mathbb{S}}'}) =$ Categorical$(\boldsymbol{X}_{\bar{\mathbb{T}}'}|\boldsymbol{p}_{\boldsymbol{\theta}}(\boldsymbol{X}_{\bar{\mathbb{S}}'}))$ is modeled as a categorical distribution with probabilities $\boldsymbol{p}_{\boldsymbol{\theta}}(\boldsymbol{X}_{\bar{\mathbb{S}}'})$, and $\boldsymbol{X}_{\bar{\mathbb{T}}}$ always contain one token (either the label $y$ or an $\boldsymbol{x}$-token). Refer also to Table 1 of the main manuscript for other details.

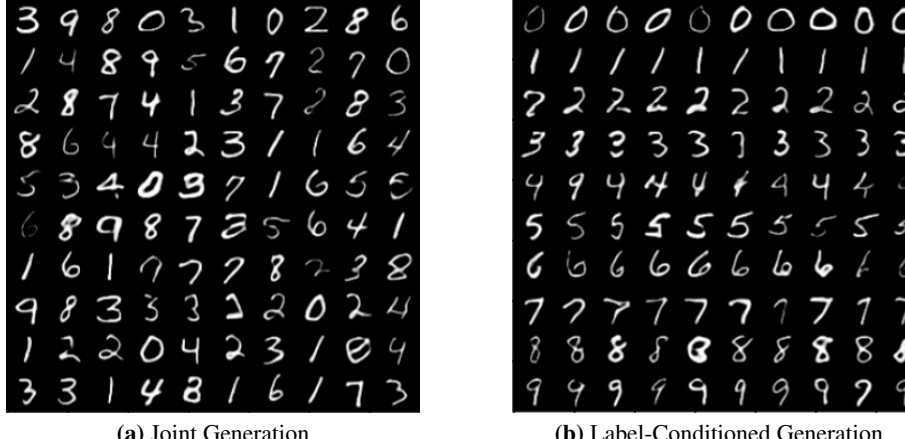

**(a)** Joint Generation  **(b)** Label-Conditioned Generation

Figure 11: Demonstration of versatile data capabilities of big learning, retrieved from $p_{\boldsymbol{\theta}}(\boldsymbol{X}_{\mathbb{T}'}|\boldsymbol{X}_{\mathbb{S}'})$ with specified $(\mathbb{S}', \mathbb{T}')$.

## F    EMPIRICAL EVALUATIONS ON THE GLUE BENCHMARK

Concerning the empirical comparisons between existing methods for foundation models and the presented big learning, intuitively, one would consider first using the big learning as the pretraining strategy in place of existing ones, followed by applying the same naïve fine-tuning on downstream tasks, to evaluate the effectiveness of the big learning. Unfortunately, we cannot afford the pretraining cost; for example, to pretrain a XLNet-Large takes about 5.5 days on **512 TPUs** according to [52]. We leave that to the community, as mentioned in the Conclusion.

To demonstrate the advantages of the big learning over existing methods for foundation models, we alternatively consider leveraging it to serve as the less expensive fine-tuning strategy. It's worth highlighting that, from another perspective, such experiments also verify the advantages of the big learning in the fields of supervised learning, when compared to existing supervised learning methods.

Specifically, we design experiments based on the Hugging Face transformers library [51], the GLUE benchmark [48], and the XLNET [52] that outperforms the BERT on many NLP tasks. We employ the same pretrained `xlnet-base-cased` model and continually train it on the downstream RTE/MRPC/SST-2 classification tasks via ($i$) the naive fine-tuning (*i.e.,* identical to the original XLNET, termed FT) and ($ii$) the big learning (termed big-learn), respectively. In other words, the pretraining phase (*i.e.,* the permutation language modeling [52], a special case of the big learning) is the same and we compare our big-learn with the naive FT during the finetuning phase.

Because the data of the downstream classification tasks contain both feature $\boldsymbol{x}$ and label $y$, we resort to the big learning settings of Section 3.3 of the main manuscript. Specifically, $\boldsymbol{X} = (\boldsymbol{y}, \boldsymbol{x})$ and the universal foundation model $p_{\boldsymbol{\theta}}(\boldsymbol{X}_{\mathbb{T}'}|\boldsymbol{X}_{\mathbb{S}'})$ has a network architecture similar to the one shown in Fig. 9 of the main manuscript. Note $p_{\boldsymbol{\theta}}(\boldsymbol{X}_{\mathbb{T}'}|\boldsymbol{X}_{\mathbb{S}'})$ consists of the pretrained XLNET backbone and a task-specific head that is attached to the output of the `<CLS>` token; for simplicity, we abuse

$\boldsymbol{\theta}$ to represent all the parameters. For a specific $(\mathbb{S}', \mathbb{T}')$ pair, $p_{\boldsymbol{\theta}}(\boldsymbol{X}_{\mathbb{T}'}|\boldsymbol{X}_{\mathbb{S}'})$ recovers $p_{\boldsymbol{\theta}}(y|\boldsymbol{x})$, *i.e.,* a conventional classifier.

With the above notations, we next formalize the objective for both FT and our big-learn.

- **FT.** Often a cross-entropy is employed, which is identical to

$$\mathcal{L}_{\text{FT}}(\boldsymbol{\theta}) = \mathbb{E}_{q_{\text{downstream}}(\boldsymbol{x},y)}[-\log p_{\boldsymbol{\theta}}(y|\boldsymbol{x})], \tag{20}$$

  where $q_{\text{downstream}}(\boldsymbol{x}, y)$ represents the training data of the downstream classification task.

- **Big-learn.** For direct comparisons, we formalize the big-learn objective as

$$\mathcal{L}_{\text{big-learn}}(\boldsymbol{\theta}) = \mathcal{L}_{\text{FT}}(\boldsymbol{\theta}) + \beta_{\text{BigLearn}}\mathcal{L}(\boldsymbol{\theta}), \tag{21}$$

  where $\beta_{\text{BigLearn}}$ is a hyperparameter and

$$\mathcal{L}(\boldsymbol{\theta}) = \mathbb{E}_{q(\mathbb{S}',\mathbb{T}')}\mathbb{E}_{q_{\text{downstream}}(\boldsymbol{X})}[-\log p_{\boldsymbol{\theta}}(\boldsymbol{X}_{\mathbb{T}'}|\boldsymbol{X}_{\mathbb{S}'})], \tag{22}$$

  with $q(\mathbb{S}', \mathbb{T}')$ denoting the sampling process of $(\mathbb{S}', \mathbb{T}')$. We simply reuse the same sampling process in Table 3.

Note Eq. (22) is equivalent to minimizing $\mathbb{E}_{q(\mathbb{S}',\mathbb{T}')}\text{KL}[q_{\text{downstream}}(\boldsymbol{X}_{\mathbb{T}'}|\boldsymbol{X}_{\mathbb{S}'})||p_{\boldsymbol{\theta}}(\boldsymbol{X}_{\mathbb{T}'}|\boldsymbol{X}_{\mathbb{S}'})]$ by referring to (1) of the main manuscript.

Table 4: Tested hyperparameters when comparing FT with big-learn on the GLUE benchmark.

| Task\Hyperparameter | Learning Rate | #Epochs | WarmUp Steps | $\beta_{\text{BigLearn}}$ |
|---|---|---|---|---|
| RTE | [2e-5, 4e-5, 6e-5] | [3, 4, 7, 10, 15] | [0, 120] | [0., 0.2, 0.4, 0.6, 0.8] |
| MRPC | [2e-5, 4e-5, 6e-5] | [3, 4, 7, 10, 15] | [0, 120] | [0., 0.2, 0.4, 0.6, 0.8] |
| SST-2 | [2e-5, 4e-5, 6e-5] | [2, 3, 4] | [0, 1200] | [0., 0.2, 0.4] |

We extensively compare FT with big-learn on the downstream RTE/MRPC/SST-2 classification tasks, by evaluating the accuracy and/or F1 score on the Dev set across the combinations of the tested hyperparameters shown in Table 4. The hyperparameters are chosen following [12; 52].

The best/median metrics are summarized in Table 2 and Fig. 12 shows the corresponding boxplots; it's clear that our big-learn consistently outperforms FT. Accordingly, the big learning can serve as a superior fine-tuning strategy. It's worth highlighting we did not carefully tune our big-learn; therefore, it's likely that its performance could be further improved by *e.g.,* tuning the sampling process $q(\mathbb{S}', \mathbb{T}')$.

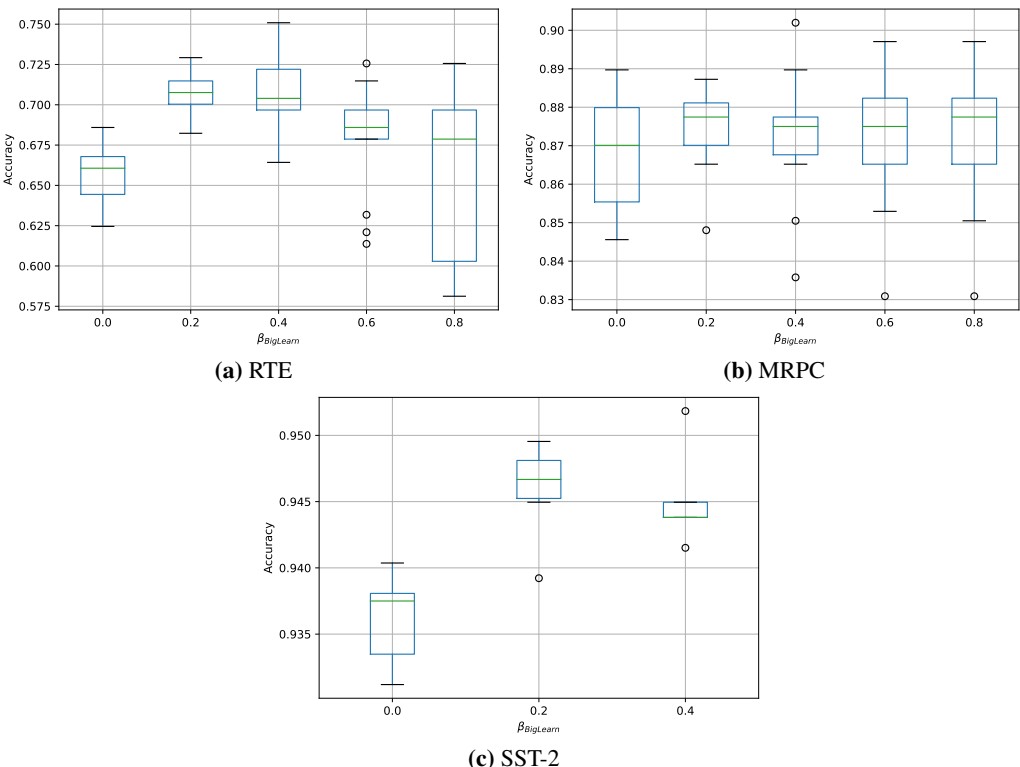

**(a)** RTE

**(b)** MRPC

**(c)** SST-2

Figure 12: Boxplots of the Dev-set accuracies from FT and our big-learn. Note big-learn with $\beta_{\text{BigLearn}} = 0$ is identical to FT (see (21)). It's clear that big-learn consistently outperforms FT on all three tasks.

We'd like to emphasize that the big learning can reduce the pretrain-finetuning gap because

- it can act as the pretraining and finetuning objectives, simultaneously;

- one can even rely on the big learning to completely merge the pretraining and finetuning phases, leading to a zero gap.

Motivated by the performance boost from the BERT to the XLNET and our discussions "on the generalization of model parameters and latent features" of Section 3.2 of the main manuscript, we posit that the big learning can serve as better pretraining and finetuning strategies than existing methods, leading to a universal machine learning paradigm. We leave the corresponding verification as future research.

## G  ADDITIONAL EXPERIMENTAL RESULTS

More experimental results, complementing the limited demonstrations of the main manuscript, are given below. Please refer to the captions for details.

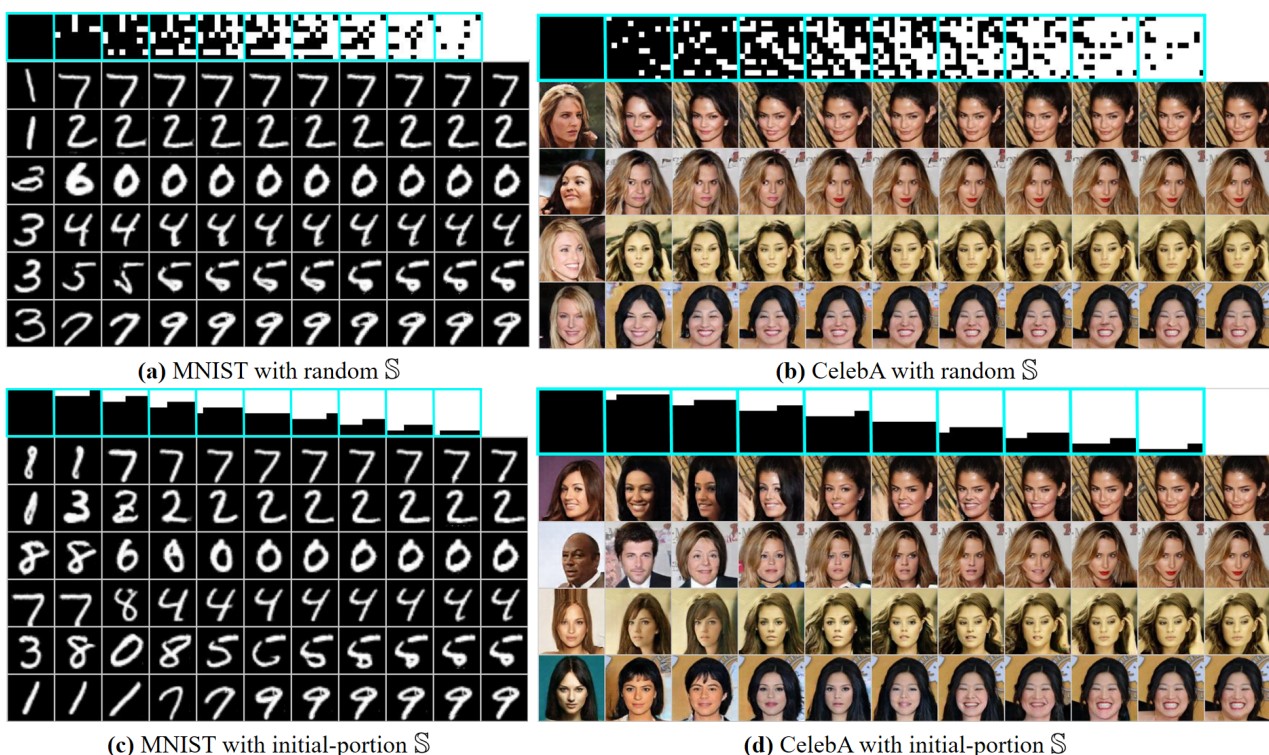

(a) MNIST with random $\mathbb{S}$  (b) CelebA with random $\mathbb{S}$

(c) MNIST with initial-portion $\mathbb{S}$  (d) CelebA with initial-portion $\mathbb{S}$

Figure 13: Demonstrating the generation/completion capabilities of big learning when gradually increasing the ratio of $\mathbb{S}$ from 0 (joint generation) to 0.9, from left to right. Shown in the light-blue boxes of the first row are the masks of $x_{\mathbb{S}}$ applied in each column; white/black indicates $\mathbb{S}/\mathbb{T}$. The right-most column shows ground-truth $x$ shared in each row. Note each row also employs the same noise. It's clear that the generations become increasingly similar/dissimilar to the ground-truth $x$ as the ratio of $\mathbb{S}$ increases/decreases, as expected. See the category, style, and thickness of the MNIST generations as the ratio of $\mathbb{S}$ decreases, as well as the identity, expression, hairstyle, and gender of the CelebA generations. Big learning produces realistic and diverse generations/completions in all situations.

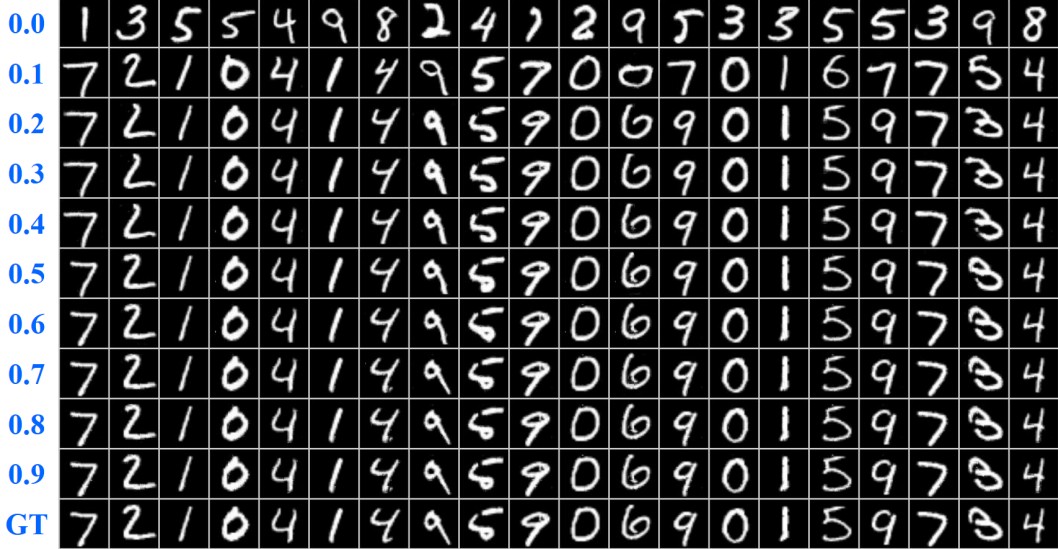

Figure 14: More MNIST generations/completions from big learning when gradually increasing the ratio of $\mathbb{S}$ from 0.0 to 0.9.

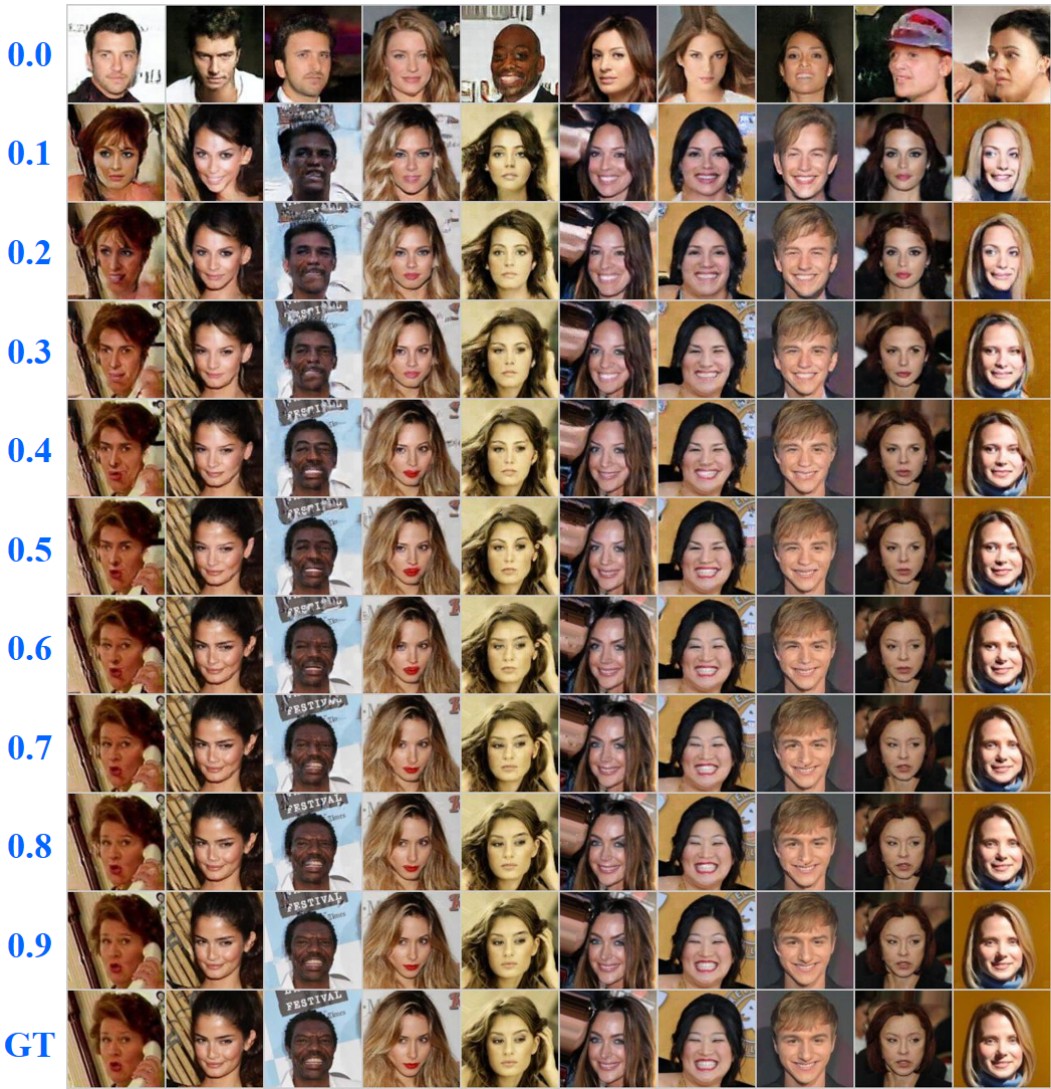

Figure 15: More CelebA generations/completions from big learning when gradually increasing the ratio of $\mathbb{S}$ from $0.0$ to $0.9$.

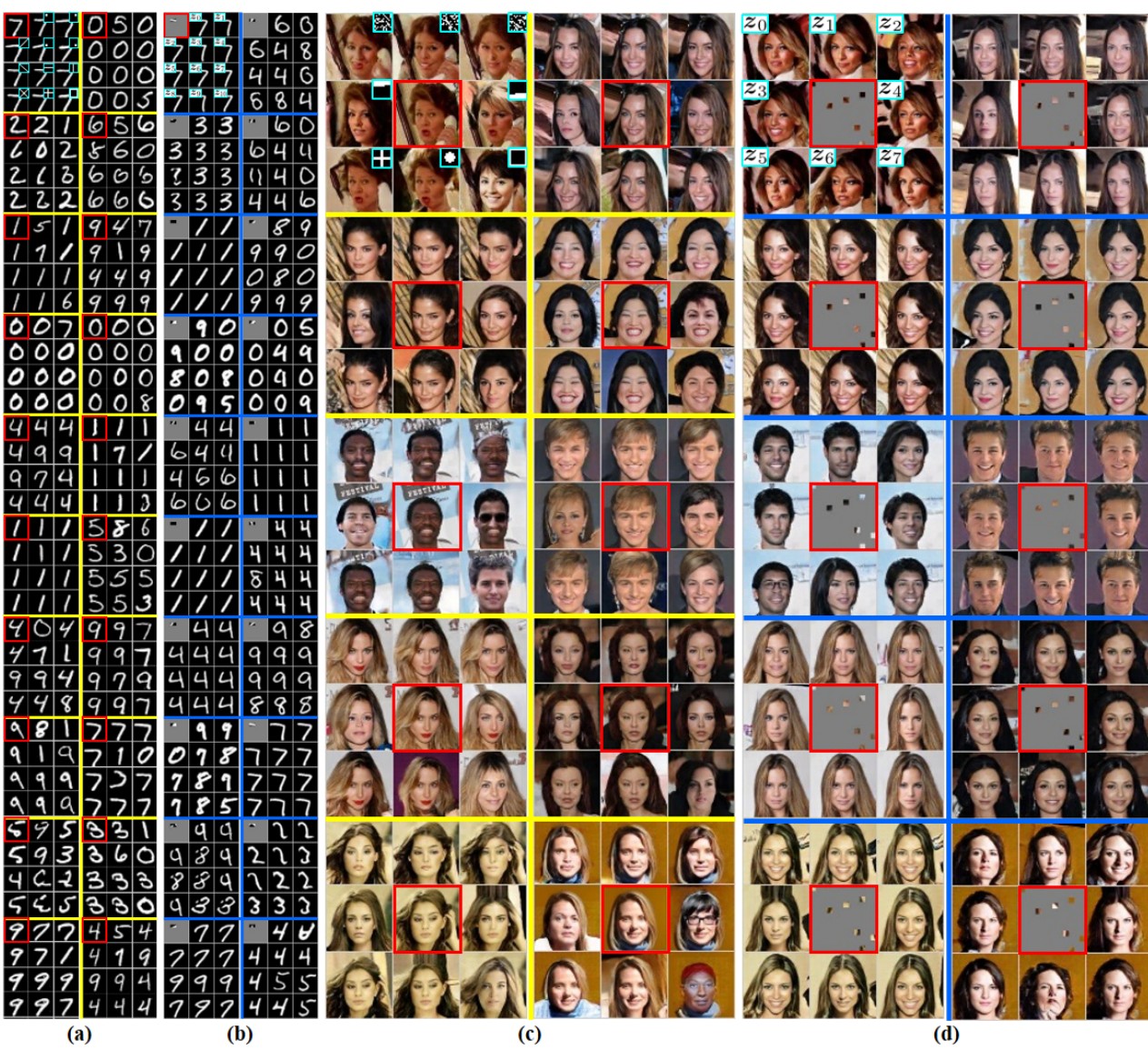

Figure 16: The diverse generations/completions of big learning with (a)(c) various $\mathbb{S}$ settings and (b)(d) different noises. Shown in red boxes are either the ground-truth images $x$ or the source $x_{\mathbb{S}}$. Big learning delivers diverse realistic generations *w.r.t.* different $\mathbb{S}$/noise settings.

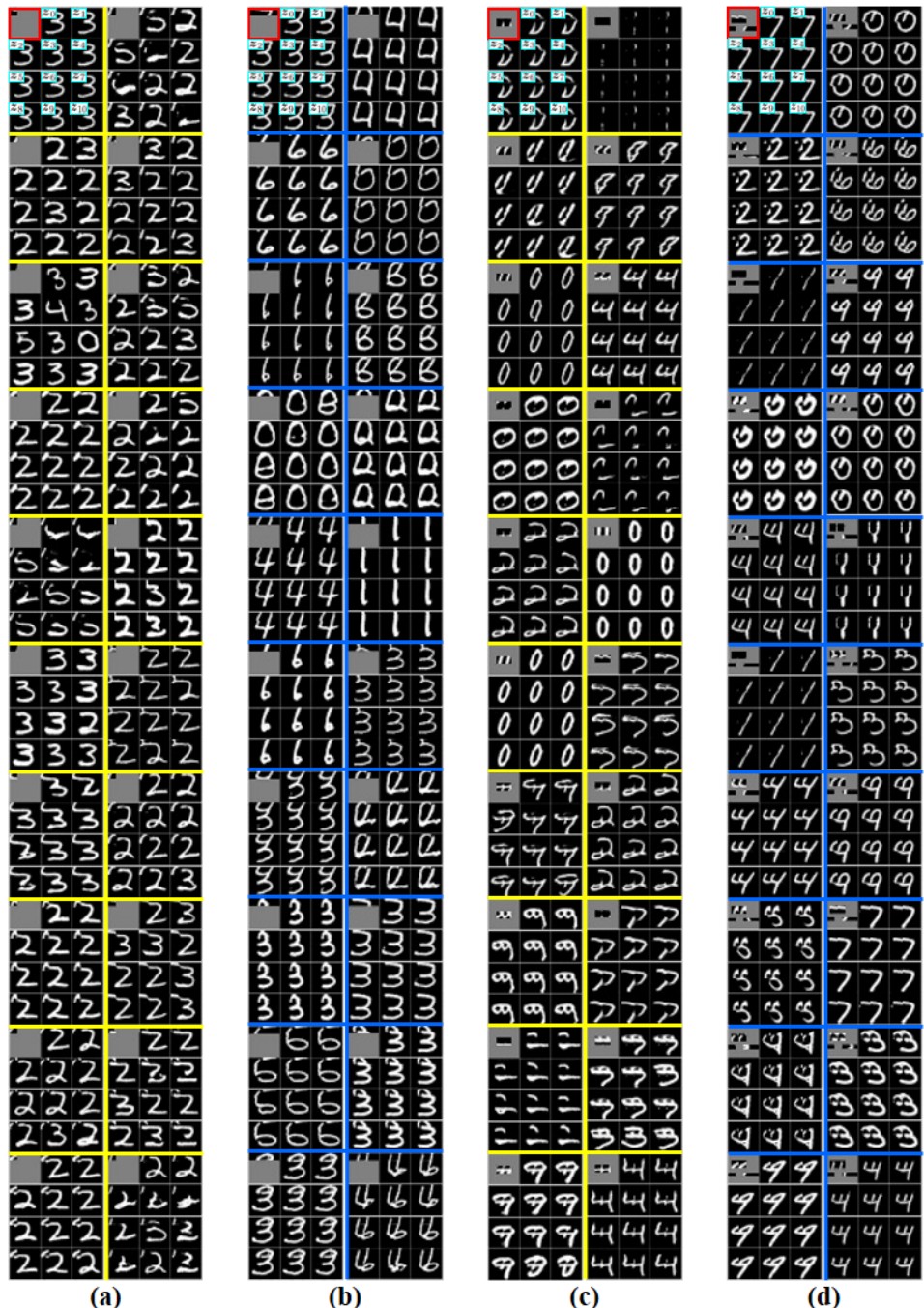

Figure 17: The strong generalization capability of big learning *w.r.t.* anomalous testing cases out of the training domain. Big learning generalizes well on $x_{\mathbb{S}}$s that are constructed with (a) random center patches replaced in the upper-left corner, (b) random center patches replaced in the upper part, (c) random center patches duplicated and replaced in the center, and (d) random patches and more complicated manipulations (including duplication, relocation, and mix-up).

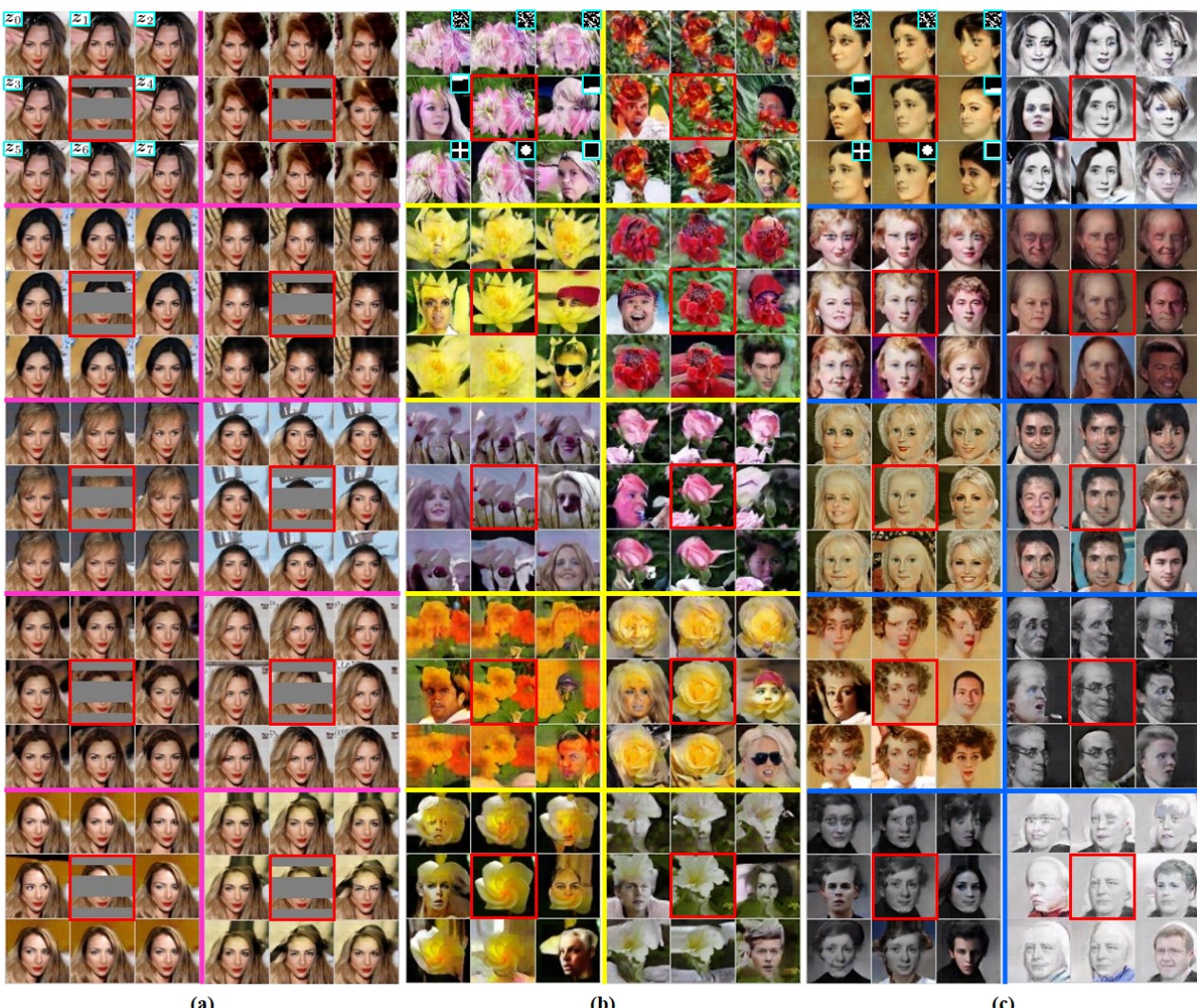

Figure 18: The strong generalization capability of big learning *w.r.t.* anomalous/unseen testing cases out of the training domain, on (a) CelebA, (b) Flowers, and (c) MetFaces. Big learning generalizes well on $x_{\mathbb{S}}$ constructed by (a) mixing-up patches from different CelebA images, (b) sampling out-of-domain image patches from the Flowers dataset, and (c) sampling out-of-domain image patches from the MetFaces dataset.

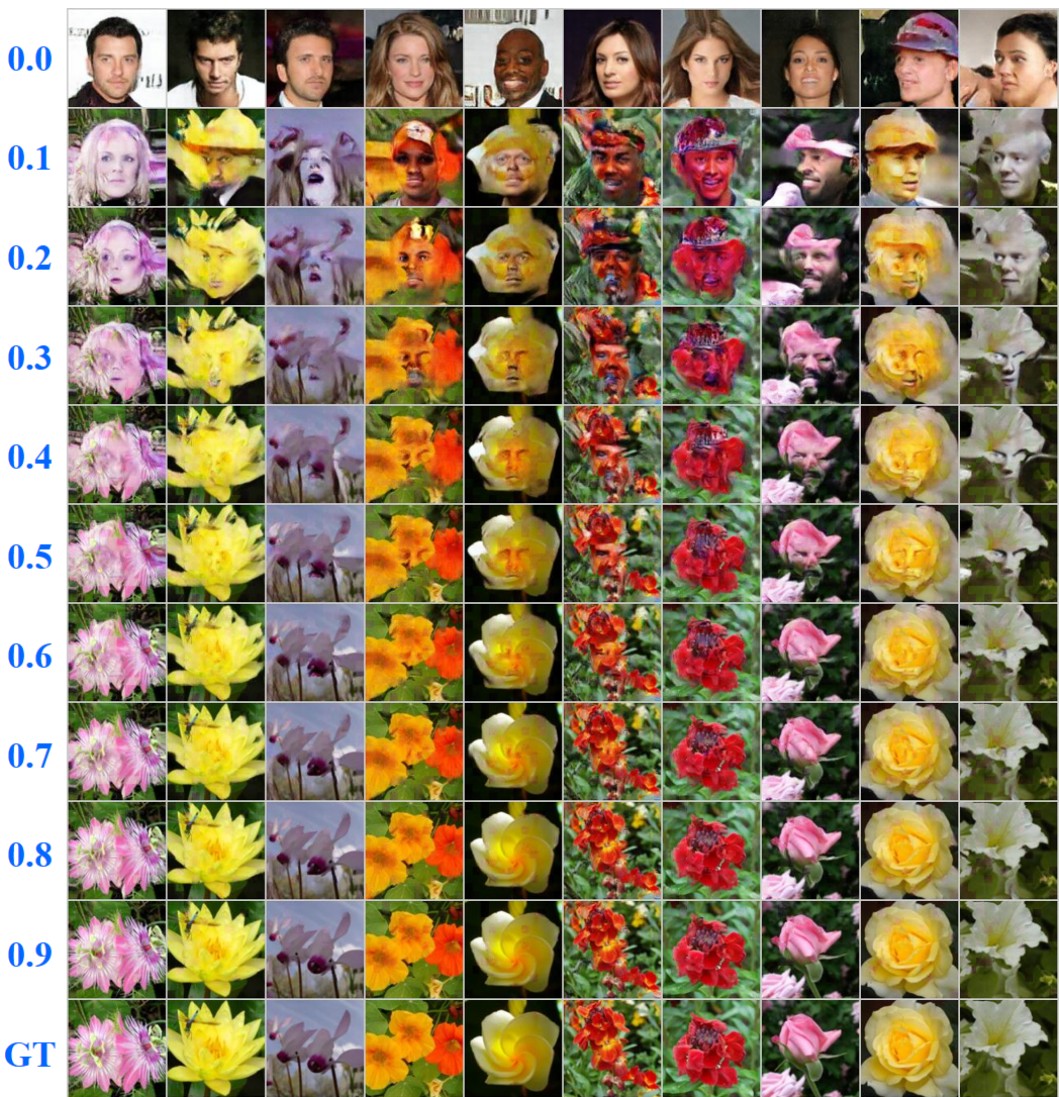

Figure 19: Out-of-domain generations/completions from big learning on the Flowers, when gradually increasing the ratio of $\mathbb{S}$ from $0.0$ to $0.9$. The tested model is big-learned on the CelebA.

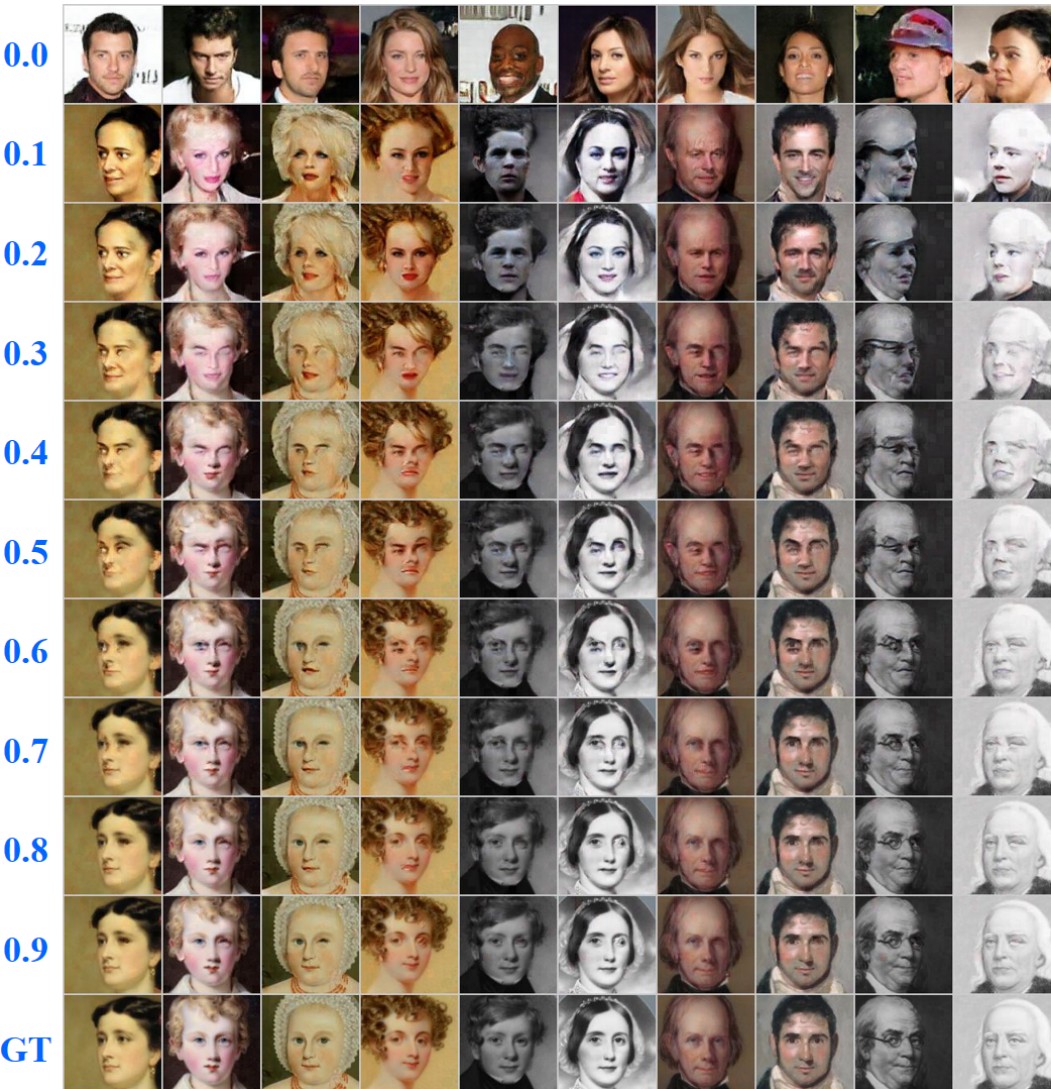

Figure 20: Out-of-domain generations/completions from big learning on the MetFaces, when gradually increasing the ratio of $\mathbb{S}$ from $0.0$ to $0.9$. The tested model is big-learned on the CelebA.

