# OpenReview forum: "Big Learning"
_ICLR.cc/2024/Conference — Submitted to ICLR 2024_

### Official Review · Reviewer_1dVT · 2023-10-30

**Soundness:** 3 good
**Presentation:** 3 good
**Contribution:** 2 fair
**Rating:** 5
**Confidence:** 3

**Summary:**

The paper introduces a so-called ‘big learning’ framework that aims to unify the objective functions of most foundation models (such as masked or AR LM or MAEs), and that learns to sample from all conditional and marginal distributions of interest. The paper suggests a big-learning variant of GANs and demonstrate for example qualitatively its performance for versatile image completion. On a GLUE benchmark, it outperforms a naïve fine-tuning strategy.

**Strengths:**

It is interesting to see that different foundation models can be unified via the big learning framework.

Learning flexible conditional/marginal generation with the suggested GAN model is new, as far as I am aware. The corresponding (non-naive) adversarial objective function is also well motivated, which I think can be seen as Kolmogorov consistency requirement.

The submission also includes qualitative results demonstrating the performance of the methods for different image competition data sets.

The paper is largely well written.

**Weaknesses:**

I am not sure how novel the approach to deliver many/all joint/conditional/marginal sampling capabilities actually is. In particular, conditional [1] or latent [2] Neural Processes models appear to address the same issue. These frameworks also use self-attention [3] or transformer [4] models. It remains unclear if the considered multi-mode training objective considered in this submission are a better choice compared to these prior works. In particular, neural process models are often also quantitatively evaluated on image impainting tasks similarly to Sec. 4.1.

Any quantitative evaluation (and in relation to prior works, e.g. [5]) of the BigLearn-GAN model seems missing. The submission claims that BigLearn-GAN yields ‘generation/completion capabilities with learned adaptive generation diversity’. Since GANs often suffer from mode collapse, it is not clear to me how their approach avoids this issue to achieve diverse sample generation.

Multimodal generative models (VAEs, etc.) have often been used to ‘unify classification and generation’ as in Sect. 4.3 and it remains unclear to me if the suggested approach improves such approaches (using common quantitative multi-modal evaluation measures).


[1] Garnelo, Marta, et al. "Conditional neural processes." International conference on machine learning. PMLR, 2018.
[2] Garnelo, Marta, et al. "Neural processes." arXiv preprint arXiv:1807.01622 (2018).
[3] Kim, Hyunjik, et al. "Attentive neural processes." arXiv preprint arXiv:1901.05761 (2019).
[4] Nguyen, Tung, and Aditya Grover. "Transformer neural processes: Uncertainty-aware meta learning via sequence modeling." arXiv preprint arXiv:2207.04179 (2022).
[5] Shih, Andy, Dorsa Sadigh, and Stefano Ermon. "Training and Inference on Any-Order Autoregressive Models the Right Way." Advances in Neural Information Processing Systems 35 (2022): 2762-2775.

**Questions:**

It would be interesting to see how the objective (6) with the 'communication' terms improve relative to the more naïve approach in (5).

Why does the objective for the cross-entropy loss assume that y given x is sampled from a categorical distribution? Why should this not work for more general $p_\theta(y|x)$?

---

> ### Author Response · Authors · 2023-11-19
> **Thanks for your comment!**
>
> **Q1: I am not sure how novel the approach to deliver many/all joint/conditional/marginal sampling capabilities actually is ... Neural Processes models appear to address the same issue...**
>
> Thank you for the constructive comments on the Kolmogorov consistency and neural processes; we will discuss the connections in the revised paper.
>
> *Comparisons between the big learning and neural processes [1-4].*
> A neural process (NP) learns a context-dependent predictive distribution $p(x_{m+1:N}|i_{m+1:N},C)$, where, e.g., $x_{m+1:N}$ are the target pixel values, $i_{m+1:N}$ are the target pixel indexes, and the context $C=(x_{1:m},i_{1:m})$ contains $m$ source pixels.
> Therefore,
> - NPs focus on diverse conditional distributions and deliver conditional sampling capabilities, whereas the more general big learning models many/all joint, conditional, and marginal data distributions and delivers joint, conditional, and marginal sampling capabilities simultaneously;
> - NPs are trained similar to a VAE within the maximum likelihood (ML) learning paradigm, while the big learning is significantly more flexible, as it can be tailored to incorporate e.g., VAEs, EMs, GANs, diffusion models, etc;
> - NPs are often constructed in the pixel level, while the big learning is proposed for handling image patches.
>
> To summarize, the big learning is more general, has superior modeling power, is more flexible wrt the tailored training algorithm, and aims at more complicated real-world applications associated with foundation models (FMs). Actually, should the big learning be generalized with Implicit Neural Representation, one may more or less interpret NPs as special cases of the generalized big learning.

---

> ### Author Response · Authors · 2023-11-19
>
> **Q2: Any quantitative evaluation (and in relation to prior works, e.g. [5]) of the BigLearn-GAN model seems missing.**
>
> Generally speaking, the MAC [5], a modified variant of the Permutation LM (see Eqs. (3) and (7) of [5]), *is a special case* of the big learning within the ML learning paradigm, because the ultimate goal of the MAC is $p_{\theta}(x_i|x_e) \rightarrow q(x_i|x_e)$, for arbitrary edge (univariate conditional) distributions.
> Accordingly, the big learning and its special case of the MAC have the same quantitative performance.
>
> Considering the comparisons between the likelihood-based MAC [5] and the GAN-based BigLearn-GAN, it's not easy to quantitatively compare both methods, because the MAC is evaluated with pixel-level test likelihoods but those likelihoods are intractable for the patch-based BigLearn-GAN.
> Due to the time constraint of the rebuttal, we provide the following analysis to address your concerns.
> - Often GANs have better generation performance than ML-based models like a flow or a PixelCNN++. Based on Tables 3-5 of [5], the likelihood-based MAC has comparable performance with PixelCNN++/ACFlow.
> - The pixel-level MAC is evaluated on image of size $32\times 32$, while the BigLearn-GAN is demonstrated with more challenging $120\times 120$ CelebA images.
> - Furthermore, by comparing the demonstrated images in the appendixes of both papers, one can observe a clear superior image fidelity from the proposed BigLearn-GAN.

---

> ### Author Response · Authors · 2023-11-19
>
> **Q3: The submission claims that BigLearn-GAN yields ‘generation/completion capabilities with learned adaptive generation diversity’. Since GANs often suffer from mode collapse, it is not clear to me how their approach avoids this issue to achieve diverse sample generation.**
>
>
> Many techniques have been proposed to successfully alleviate the notorious unstable training and mode collapse of GANs.
> We employed the gradient penalty technique (Mescheder et al., 2018) to alleviate both issues.
> On the more challenging CelebA datasets, we employed several additional techniques (detailed in Appendix D), which work reasonably well on alleviating the unstable training and mode collapse issues.
>
> Once we stabilized the training of the BigLearn-GAN, we find that the big learning naturally delivers versatile joint, conditional, and marginal sampling capabilities with *automatically learned adaptive generation diversity*. We hypothesize that the consistency and implicit regularizations among big-learning tasks are the driving force, because
> - the adaptive generation diversity is inherit in the training data (e.g., consider the target image diversity when given $90$% source patches and the diversity when given $10$% patches), and
> - the big learning uses one universal model to simultaneously model the data distribution from diverse joint, conditional, marginal perspectives, which implicitly regularize each other and force the universal model to comprehensively match the data distribution, manifested as versatile data sampling capabilities with automatically learned adaptive generation diversity.
>
> [1] L. Mescheder, A. Geiger, and S. Nowozin. Which training methods for GANs do actually converge? In ICML, pp. 3478–3487, 2018.

---

> ### Author Response · Authors · 2023-11-19
>
> **Q4: Multimodal generative models (VAEs, etc.) have often been used to ‘unify classification and generation’ as in Sect. 4.3 and it remains unclear to me if the suggested approach improves such approaches (using common quantitative multi-modal evaluation measures).**
>
> After vector-quantifying an image into discrete tokens (i.e., a sequence of "words") and considering the discrete label $y$ as a special "word", the algorithmic implementations in Secs 4.3 and 4.4 are very similar. That is, the unified classification and generation in Sec. 4.3 is algorithmically the same as the big learning strategy in Sec. 4.4; the classification in Sec. 4.3 is algorithmically the same as the naive fine-tuning strategy in Sec. 4.4.
> Accordingly, Sec. 4.4 shows that unifying classification and generation could outperform the naive classification.
> More quantitative experiments will be conducted in the future.
>
> Regarding "common quantitative multi-modal evaluation measures," see also the second item of the general response 1 on "systematic quantitative comparisons with/among existing/SOTA foundation models." The concurrent research (Bao et al., 2023) (clearly not our contribution) has extensively demonstrated that the quantitative results (e.g., the FID and CLIP score) of a big-learned UniDiffuser "are not only superior to existing general-purpose models but also comparable to the bespoken models (e.g., Stable Diffusion and DALL·E2) in representative tasks (e.g., text-to-image generation)," justifying the superiorty of the big learning principle.
> Alternatively, one may leverage the big learning to uncover the underlying learning principle of the UniDiffuser for better understanding/analysis.
>
> [2] Fan Bao, Shen Nie, Kaiwen Xue, Chongxuan Li, Shi Pu, Yaole Wang, Gang Yue, Yue Cao, Hang Su, and Jun Zhu. One transformer fts all distributions in multi-modal diffusion at scale. arXiv preprint arXiv:2303.06555, 2023.

---

> ### Author Response · Authors · 2023-11-19
>
> **Q5: It would be interesting to see how the objective (6) with the 'communication' terms improve relative to the more naïve approach in (5).**
>
> Thank you for the suggestion. Due to the time constraint, we will work on it after the rebuttal.
>
> **Q6: Why does the objective for the cross-entropy loss assume that y given x is sampled from a categorical distribution? Why should this not work for more general $p_{\theta}(y|x)$?**
>
> Assume $\boldsymbol{y} \in (0,1)^C$ is a one-hot vector (often denoting a discrete label) and a probability vector $\boldsymbol{P} \in R_{+}^C$ with $\sum_{c=1}^{C} P_c=1$ (often $\boldsymbol{P}$ is a neural-network-parameterized function of an input image $\boldsymbol{x}$).
> Then, the log-likelihood of a categorical distribution, i.e., $\log p_{\theta}(y|\boldsymbol{x}) = \log Cat(\boldsymbol{y}|\boldsymbol{P})=\log\prod\nolimits_{c=1}^{C} P_c^{y_c} = \sum\nolimits_{c=1}^{C} y_c \log P_c$, is identical to the negative cross-entropy loss.
> Accordingly, minimizing the cross-entropy loss can be explained by maximum log-likelihood learning with a categorical assumption.
> The categorical distribution is popularly used for modeling a discrete random variable (RV) like a label, whereas more general distributions for a discrete RV likely introduce more advanced losses, e.g., the focal loss.

---

### Official Review · Reviewer_dsm7 · 2023-11-01

**Soundness:** 2 fair
**Presentation:** 2 fair
**Contribution:** 2 fair
**Rating:** 5
**Confidence:** 4

**Summary:**

This paper proposes the big learning that exhaustively exploits the available data information and potentially delivers all joint, conditional, and marginal sampling data capabilities. The authors claim that the big learning (i) comes with extraordinary training flexibilities for complete/incomplete data and for customizing training tasks, (ii) contains most objectives of foundation models as special cases, and (iii) is a new dimension for upgrading conventional machine learning paradigms; Specifically, this paper presents the upgraded BigLearn-GAN as a demonstration example and experiments justify the effectiveness of the presented big learning.

**Strengths:**

The authors try to propose a big learning framework, which contains most objectives of foundation models as special cases and potentially delivers all joint, conditional, and marginal sampling data capabilities.

**Weaknesses:**

Quality/Clarity: the paper is hard to follow. The title is too big and it is hard to know its contribution since it aggregates the existing approaches and wants to put everything under this framework. And if BigLearn-GAN is the contribution, then please compare it with the state of the art.


Originality/significance: the idea is ok, which wants to put all models under this big learning framework. However, it only aggregates the current approaches, and these approaches are known and did before. And if there were a big learning framework, can you guild the machine learning research in the future (for example, design a new model/architecture)? Also I did not see any novelty here.

**Questions:**

The authors try to unify everything under big learning framework, and if there is such framework, how does it guild our future research? for example, can we find something new (either theory or experimental level) from this framework?

---

> ### Author Response · Authors · 2023-11-19
> **Thanks for your comment!**
>
> **Q1: the paper is hard to follow. The title is too big and it is hard to know its contribution...**
>
> Please see our general responses on "our main contributions" and "systematic quantitative comparisons with/among existing/SOTA foundation models." We have added new Figs. 1 and 6 to the revised paper to highlight the main motivations and big picture of the big learning to improve the clarity.
>
> We believe the proposed unified, consistent, and principled big learning framework, which *simultaneously* delivers many valuable bonuses (see the general response 2 on ``our main contributions''), will contribute significantly to the machine learning community.

---

> ### Author Response · Authors · 2023-11-19
>
> **Q2: The authors try to unify everything under big learning framework, and if there is such framework, how does it guild our future research? for example, can we find something new (either theory or experimental level) from this framework?**
>
> We consider it a solid contribution to give a positive answer to the zero-to-one question whether there is a unified and principled learning framework for foundation models (FMs).
> We answered that question by proposing the big learning and by extensively verifying its feasibility with diverse experiments.
>
> The big learning might guild future machine learning research in two directions:
> - it may serve as a high-level guidance for developing novel FM algorithms with wider scope, e.g., one may combine the big learning principle with GANs, diffusion models, or energy-based models to develop a new FM algorithm for specific applications.
> The unified big learning also offers certain potential for combining multiple FM algorithms to deal with very complicated real-world challenges and for analyzing different FM algorithms within one unified and consistent framework.
>
> - the big learning is a new orthogonal dimension for upgrading conventional machine learning paradigms. We have proved that the big learning principle can be leveraged to upgrade the conventional EM, VAE, GAN, and diffusion models into the upgraded BigLearn-EM, BigLearn-VAE, BigLearn-GAN, and BigLearn-diffusion models (not our contribution; (Bao et al., 2023)), which naturally possess significantly enhanced (joint, conditional, and marginal) data capabilities than their conventional counterparts.
>
> The proposed unified, flexible, and principled big learning framework, for the first time, enables high-level knowledge transfer from conventional machine learning to FMs, from FMs to conventional machine learning, etc.
> Combining those contributions with many other bonuses from the big learning (see the general response 2 on ``our main contributions''), we consider our contributions novel and valuable to the machine learning community.
>
> [1] Fan Bao, Shen Nie, Kaiwen Xue, Chongxuan Li, Shi Pu, Yaole Wang, Gang Yue, Yue Cao, Hang Su, and Jun Zhu. One transformer fts all distributions in multi-modal diffusion at scale. arXiv preprint arXiv:2303.06555, 2023.

---

### Official Review · Reviewer_7mKq · 2023-11-05

**Soundness:** 2 fair
**Presentation:** 1 poor
**Contribution:** 2 fair
**Rating:** 5
**Confidence:** 4

**Summary:**

In this paper, the authors propose a general problem called "Big Learning", which generalize various important problems e.g. masked/casual/autoregressive LM, supervised classification, generations,... They argue that big learning can be leveraged to deliver joint, conditional, and marginal data sampling capabilities with one universal foundation model. Then they study the generative adversarial net
(GAN) in the context of big-learning, where the variant is named the BigLearn-GAN. Experiments show the performance of BigLearn-GAN for MNIST/CelebA datasets and the GLUE benchmark.

**Strengths:**

This paper proposes to generalize a problem for masked/casual/autoregressive LM, supervised classification, generation, which could provide some structural perspective. The authors investigated the GAN network in this context and conduct experiments to test their method.

**Weaknesses:**

1. The presentation is poor, without the examples the definition of the big learning problem is unclear. The authors should define $x_T$ and $x_S$. Most of the examples only have one pair of $(T, S)$, not a collection.

2. This paper lacks of systematic quantitative comparisons between the proposed approach and existing methods on pretraining foundation models with large-scale data.
3. The claims are not sufficiently supported by the experiments: while the authors argue the potential of the learning framework, the quantitative experiments (table 2 only) is too small and cannot be representative for training FMs.

**Questions:**

It is important to compare the computational cost of the proposed approach with prior method. Can the authors discuss this?

---

> ### Author Response · Authors · 2023-11-19
> **Thanks for your comment!**
>
> **Q1: ...the definition of the big learning problem is unclear. The authors should define $x_T$ and $x_S$. Most of the examples only have one pair of $(T,S)$, not a collection.**
>
> Thank you for the constructive suggestions. In the revised paper, we have added a new Fig. 1 to the main paper and a new Fig. 6 to the Appendix to highlight
> - the definitions of $(x_T, x_S)$ and $(T,S)$ (with a collection of examples),
> - our main motivations, and
> - the big picture of the big learning,
>
> to improve the clarity.

---

> ### Author Response · Authors · 2023-11-19
>
> **Q2: This paper lacks of systematic quantitative comparisons between the proposed approach and existing methods on pretraining foundation models with large-scale data. ... quantitative experiments (table 2 only) is too small and cannot be representative for training FMs.**
>
> We argue that systematic quantitative comparisons with/among existing foundation models (FMs) are not the primary concern when proposing a unified, consistent, and principled big learning framework that contains most FMs and learning paradigms as special cases.
>
> We address your concerns by first providing detailed discussions on ``systematic quantitative comparisons with/among existing/SOTA foundation models'' (see our general response $1$), followed by highlighting our main contributions (see our general response $2$), which we believe are valuable to the machine learning community.

---

> ### Author Response · Authors · 2023-11-19
>
> **Q3: It is important to compare the computational cost of the proposed approach with prior method. Can the authors discuss this?**
>
>
> The big learning principle is a general concept that contains most FMs and many learning paradigms as special cases. Accordingly, similar to the discussions on ``systematic quantitative comparisons with/among existing/SOTA FMs,'' the big learning has the same computational cost with its special cases.
>
> Should the reviewer concern about the comparisons among different big-learning instantiations (i.e., different FM algorithms) like masked LM versus BigLearn-GAN, we highlight that the comparisons might not be necessary, because they are often used for different applications.
> Despite that, the BigLearn-GAN is likely more computationally expensive, because of its additional discriminator, regularizer, and communication loss (see the newly introduced Appendix Algorithm 2 of the revised paper). But one also receives the valuable benefits from the corresponding adversarial training and big learning.
>
>
> Considering a general analysis on the computational cost of the big learning, we have provided a new Algorithm 1 in the Appendix of the revised paper.
> In each iteration, we randomly sample **one** $(S,T)$ pair and then perform the selected matching of $p_{\theta}(x_T|x_S) \rightarrow q(x_T|x_S)$, which is similar to one step of conventional update of $p_{\theta}(x) \rightarrow q(x)$, with limited additional overhead to rearrange the data $(x_S,x_T)$ and to embed $(S,T)$.
> Therefore, the computational cost for each iteration is roughly the same.
> However, since the big learning masters many/all joint, conditional, and marginal data capabilities simultaneously during training, it needs moderately more iterations to converge. Empirically, we find that $200K$ iterations are sufficient for the big learning.

---

### Official Review · Reviewer_dok4 · 2023-11-06

**Soundness:** 3 good
**Presentation:** 2 fair
**Contribution:** 3 good
**Rating:** 6
**Confidence:** 4

**Summary:**

This paper introduces a new learning paradigm called the Big Learning, where the learning algorithms can simultaneously model joint, conditional, and marginal distributions, by exploring incomplete training data. The authors show that many existing learning paradigms can be viewed as special cases of the proposed Big Learning paradigm, e.g., Masked LM.

After rebuttal: I have read the rebuttal and would like to keep my scores.

**Strengths:**

The authors introduce a new concept call the Big Learning, which unifies many existing learning paradigms such as Mask LM. Based on this new learning concept, the author proposes advanced versions of GAN and maximum likelihood learning. The authors also run experiments and show the efficacy of the proposed methods.

**Weaknesses:**

I have mixed feelings about this paper. While the authors propose a new learning paradigm Big Learning that can unify some existing learning paradigms, it seems to me that this new learning paradigm is just a slightly more "advanced" version of self-supervised learning. Can authors highlight the main differences?

Also, the description of experiments in Section 4 is not clear to me, e.g., what is the experiment setups for the ones shown in Fig 2, 3, 4?

**Questions:**

Please see comments above.

---

> ### Author Response · Authors · 2023-11-19
> **Thanks for your comment!**
>
> **Q1: ...is just a slightly more "advanced" version of self-supervised learning. Can authors highlight the main differences?**
>
> The presented big learning condenses most learning objectives of foundation models (like masked LM, causal/auto-regressive LM, and permutation LM) and many machine learning paradigms (like adversarial learning and maximum-likelihood learning) into one consistent and unified framework in a principled way (i.e., via consistent joint, marginal, and conditional distribution matchings across potentially diverse domains).
> Please also refer to Fig. 1 and Appendix Fig. 6 of the revised paper for the big picture of the big learning.
> By comparison, existing self-supervised learning methods are somewhat heuristic.
>
> When compared with heuristic self-supervised learning methods represented by masked LM (special cases of the big learning), our big learning
> - is a more general, flexible, and principled learning framework,
> and
> - it serves as the theoretical foundation for these methods, by revealing their underlying learning principle of multi-tasking conditional matchings (see the corresponding rows of Table 1).
>
> For other self-supervised learning methods exampled by contrastive learning, please refer to our discussions on ``big learning versus self-supervised contrastive learning'' in the last paragraph before Section 4.
>
> Furthermore, we'd like to highlight that the universality, flexibility, and principledness of the big learning also provide extra valuable bonuses that can not be obtained from existing self-supervised learning methods.
> For example,
> - the big-learned model *simultaneously* delivers many/all joint, conditional, and marginal data sampling capabilities;
> - the unified and flexible big learning is better suited to complicated real-world applications with multi-modality (see Section 3.3);
> and
> - the principled big learning enables knowledge feedback from advanced foundation models to conventional machine learning paradigms (e.g., we have leveraged the big learning to upgrade the GAN/EM/VAE into BigLearn-GAN/BigLearn-EM/BigLearn-VAE).
>
>
> Accordingly, by referring to the general response 2 on ``Our main contributions,'' we believe that the big learning, condensing many learning paradigms into a consistent, unified, and principled framework, will contribute significantly to the machine learning community.

---

> ### Author Response · Authors · 2023-11-19
>
> **Q2: the description of experiments in Section 4 is not clear to me, e.g., what is the experiment setups for the ones shown in Fig 2, 3, 4?**
>
>
> The main goal of the experiments is to demonstrate the feasibility and effectiveness of the uni-modal big learning in Eq. (3) and the multi-modal big learning in Eq. (7).
>
> Figs. 2-4 (Figs. 3-5 in the revised paper) are presented to demonstrate the effectiveness of the uni-modal big learning in Eq. (3), i.e., $p_{\theta}(x_{T}|x_{S})  \longrightarrow q(x_{T}|x_{S}), \forall (S,T)$. Specifically, we instantiate the uni-modal big learning with the BigLearn-GAN with Eqs. (4-5), where the goal is to adversarially train a universal model $p_{\theta}(x_{T}|x_{S})$ to yield $p_{\theta}(x_{T}|x_{S})  \longrightarrow q(x_{T}|x_{S}), \forall (S,T)$ (i.e., many/all joint, marginal, and conditional data sampling capabilities).
>
> - $x_{S}/x_{T}$ denotes the input-source/output-target image patches, with $S/T$ being a set that contains the input/output patch indexes; see Fig. 1 and Appendix Fig. 6 of the revised paper for explicit demonstrations.
> In the experiments, the $(S,T)$ indexes are randomly sampled from the whole patch index set $L$; see Appendix A and Table 3 for the implementation details.
>
> - The model $p_{\theta}(x_{T}|x_{S})$ (i.e., the BigLearn-GAN generator) is constructed by modeling its generative process, that is, $x_{T}=g(x_{S},z), z \sim N(0,I), \forall (S, T)$, where $g(\cdot)$ is illustrated in Appendix Fig. 9(a) (Fig. 10(a) in the revised paper).
>
> - After big learning, the universal BigLearn-GAN generator, when given a fixed noise $z$, is expected to generate realistic images $x_{T \cup S}$ based on different $x_S$ with different settings for both $x$ and $S$.
> In Fig. 2 (Fig. 3 of the revised paper), $x_S$ is constructed with a test data $x$ (e.g., the last image of each row) and a user-defined $S$ (the top binary mask of each column; all patches within $S$ are colored white; e.g., $S=\emptyset$ in the first column).
> The sub-figure (a)/(b) demonstrates the generation power with randomly-selected/initial-portion $S$s, with an increasing $S$-ratio from the left to the right.
>
> - The universal BigLearn-GAN generator with $x_{T}=g(x_{S},z), z \sim N(0,I), \forall (S, T)$ is also expected to generate realistic images with both user-controlled $S$ and random noise $z$. In both sub-figures (a) and (b) of Fig. 3 (Figs. 4 of the revised paper), the left half shows the realistic generation with diverse $S$s, whereas the right half shows the diverse generation with different noise $z$ conditioned on the same $x_S$.
>
> - Since the BigLearn-GAN generator enables flexible inputs, we challenge its capability with abused $x_{S}$ in Fig.4 (Fig. 5 of the revised paper), e.g., $x_{S}$ constructed by relocating and/or duplicating the center patches in sub-figure (a), $x_{S}$ combining patches from different images in sub-figure (b), and out-of-domain $x_S$ in sub-figure (c).
>
> Sections 4.3 and 4.4 are presented to demonstrate the effectiveness of the multi-modal big learning in Eq. (7), where we focused on discrete observations and resorted to the maximum-likelihood instantiation in Section 3.2.2.
> Section 4.3 revealed the feasibility of leveraging the big learning to unify classification and generation, and Section 4.4 demonstrated that the big learning, combining generation with classification, is capable of serving as a superior fine-tuning strategy than the naive one.

---

### Author Response · Authors · 2023-11-19
**General Response**

We appreciate the efforts and constructive comments every reviewer made in reviewing our submission. The revised paper for this stage has been uploaded, with new figures and algorithms provided to demonstrate the big picture and implementation details of the big learning, for improved clarity.
Most discussions during the rebuttal will be incorporated into the next revision of the paper.

Below we first respond to 2 main comments; the rest ones are separately addressed under the relevant review.
We're looking forward to hearing your comments. Thank you again for your feedback and consideration.

---

> ### Author Response · Authors · 2023-11-19
> **General Response 1: Systematic quantitative comparisons with/among existing/SOTA foundation models**
>
> Systematic quantitative comparisons with/among existing/SOTA foundation models (FMs) are not the primary concern when proposing a unified, consistent, and principled big learning framework that contains most FMs and learning paradigms as special cases.
>
> $(i)$ Since most foundation models (FMs) are special cases of the big learning, there is no need to quantitatively compare them, because they are the same in the corresponding settings.
> Those successful FMs are strong evidences for the effectiveness of the big learning, from a variety of special-case perspectives.
>
> $(ii)$ **Can the big learning principle be tailored to produce a better FM training algorithm? Yes.** Due to the time constraint of the rebuttal, we refer to a concurrent research (Bao et al., 2023) (clearly not our contribution) to address the concern.
> Generally, Bao et al. (2023) leveraged the big learning principle (in all the transformed/noised latent spaces) to upgrade the conventional diffusion model into a BigLearn-diffusion model termed the UniDiffuser, whose ``quantitative results (e.g., the FID and CLIP score) are not only superior to existing general-purpose models but also comparable to the bespoken models (e.g., Stable Diffusion and DALL·E2) in representative tasks (e.g., text-to-image generation),'' justifying the superiorty of the big learning principle.
> One may also leverage the big learning to uncover the underlying learning principle of the UniDiffuser for better understanding/analysis.
>
> $(iii)$ To enable systematic quantitative comparisons among FMs and, more importantly, to analyze the reasons why one FM performs better, we will need a theoretical platform/framework where most FMs can be analyzed in a unified, consistent, and principled manner. The proposed big learning serves as a competitive candidate.
>
> $(iv)$ Last but not least, to systematically and quantitatively compare FMs based on the aforementioned platform is prohibitively expensive, considering the general purpose of FMs and their massive downstream tasks.
> Only industry giants might have the computational resource needed.
> We hope they will spend their resource to make systematic quantitative comparisons among FMs in the future, where the big learning principle is expected to contribute to a large-scale unified, consistent, and principled testing environment.
>
> [1] Fan Bao, Shen Nie, Kaiwen Xue, Chongxuan Li, Shi Pu, Yaole Wang, Gang Yue, Yue Cao, Hang Su, and Jun Zhu. One transformer fts all distributions in multi-modal diffusion at scale. arXiv preprint arXiv:2303.06555, 2023.

---

> ### Author Response · Authors · 2023-11-19
> **General Response 2: Our main contributions**
>
> $(i)$ Foundation models (FMs) are the most powerful driving force for AI. However, most of their training algorithms are somewhat heuristic, such as masked LM (i.e., mask-and-predict), causal LM, and permutation LM. The rapid development of FMs and the analysis/explanation of their trustworthiness require a unified and principled theoretical framework. The proposed big learning serves as a competitive candidate.
>
> $(ii)$ To give a positive answer to the zero-to-one question ``whether there is a unified and principled learning framework for FMs'' is significantly valuable to the machine learning community.
> We answered that question by proposing the big learning principle and by extensively verifying its feasibility with diverse experiments.
>
> $(iii)$ Instead of making prohibitive systematic quantitative comparisons with existing/SOTA FMs, we have shown that the principled big learning enables knowledge feedback from advanced FMs to conventional machine learning paradigms, e.g., we have used the big learning principle to upgrade the GAN/EM/VAE into BigLearn-GAN/BigLearn-EM/BigLearn-VAE (the BigLearn-GAN is presented in this paper, while the BigLearn-EM and BigLearn-VAE are given in separated papers in the supplementary materials).
> Moreover, the aforementioned concurrent research (Bao et al., 2023) (clearly not our contribution) implicitly leveraged the big learning principle to upgrade a diffusion model into a BigLearn-diffusion model, which demonstrated SOTA performance on multiple diverse large-scale text-image experiments with one universal model.
> Therefore, *the big learning is a novel FM-inspired dimensionality for upgrading conventional machine learning paradigms.*
>
> $(iv)$ To summarize, we have presented the unified, consistent, and principled big learning framework for FMs and machine learning, which *simultaneously*
>
> - delivers a theoretical framework revealing what most FMs are implicitly doing; e.g., masked LM does multiple conditional matchings based on the mean-field assumption (see Table 1);
>
> - generalizes existing FM training algorithms to enable training a universal model to *simultaneously* deliver many/all joint, conditional, and marginal data sampling capabilities;
>
> - unifies and generalizes FM training algorithms in situations with complete/incomplete data and uni-modal/multi-modal data modality; and
>
> - enables knowledge feedback from advanced FMs to conventional machine learning paradigms like GAN, EM, VAE, diffusion models, \etc and vise versa.
>
> We believe such a general framework, which *simultaneously* delivers all the above-mentioned bonuses, is valuable to the machine learning community.
>
>
> [1] Fan Bao, Shen Nie, Kaiwen Xue, Chongxuan Li, Shi Pu, Yaole Wang, Gang Yue, Yue Cao, Hang Su, and Jun Zhu. One transformer fts all distributions in multi-modal diffusion at scale. arXiv preprint arXiv:2303.06555, 2023.

---

### Meta-Review · Area_Chair_dayy · 2023-12-24

**Metareview:**

The paper proposed a big learning framework which tries to unify many machine learning problems including many self-supervised learning problems and many generative models. The paper used big-learn-GAN as a concrete application of the framework. While reviewers appreciate the idea of having a unified framework, most reviewers raise concerns on the novelty and impact of the framework. There is only one concrete new application big-learn-GAN, and yet its superiority over SOTA generative models is not clear. I think the authors need to add more results or experiments to justify the power of the new framework. I recommend reject.

**Justification For Why Not Higher Score:**

Most reviewers find there is not enough justification on the power of the proposed framework.

**Justification For Why Not Lower Score:**

N/A

---

### Decision · Program_Chairs · 2024-01-16

Reject